# As large as it gets – Studying Infinitely Large Convolutions via Neural Implicit Frequency Filters

**Julia Grabinski**                                                            *julia.grabinski@uni-mannheim.de*
*Chair of Machine Learning*
*University of Mannheim*

**Janis Keuper**                                                                              *keuper@imla.ai*
*Institute for Machine Learning and Analytics (IMLA)*
*University of Offenburg*

**Margret Keuper**                                                          *margret.keuper@uni-mannheim.de*
*Chair of Machine Learning*
*University of Mannheim and*
*Max Planck Institute for Informatics*
*Saarland Informatics Campus, Germany*

**Reviewed on OpenReview:** *https://openreview.net/forum?id=xRy1YRcHWj*

## Abstract

Recent work in neural networks for image classification has seen a strong tendency towards increasing the spatial context during encoding. Whether achieved through large convolution kernels or self-attention, models scale poorly with the increased spatial context, such that the improved model accuracy often comes at significant costs. In this paper, we propose a module for studying the effective filter size of convolutional neural networks (CNNs). To facilitate such a study, several challenges need to be addressed: (i) we need an effective means to train models with large filters (potentially as large as the input data) without increasing the number of learnable parameters, (ii) the employed convolution operation should be a plug-and-play module that can replace conventional convolutions in a CNN and allow for an efficient implementation in current frameworks, (iii) the study of filter sizes has to be decoupled from other aspects such as the network width or the number of learnable parameters, and (iv) the cost of the convolution operation itself has to remain manageable i.e. we can not naïvely increase the size of the convolution kernel. To address these challenges, we propose to learn the *frequency representations* of filter weights as neural implicit functions, such that the better scalability of the convolution in the frequency domain can be leveraged. Additionally, due to the implementation of the proposed neural implicit function, even large and expressive spatial filters can be parameterized by only a few learnable weights. Interestingly, our analysis shows that, although the proposed networks could learn very large convolution kernels, the learned filters are well localized and relatively small in practice when transformed from the frequency to the spatial domain. We anticipate that our analysis of individually optimized filter sizes will allow for more efficient, yet effective, models in the future.

## 1 Introduction

Recent progress in image classification such as Liu et al. (2022b) builds upon observations on the behavior of vision transformers Dosovitskiy et al. (2020); Khan et al. (2022); Touvron et al. (2021); Vaswani et al. (2017), which rely on the learned self-attention between large image patches and therefore allow information from large spatial contexts to be encoded. In particular, we have recently seen a trend towards increasing

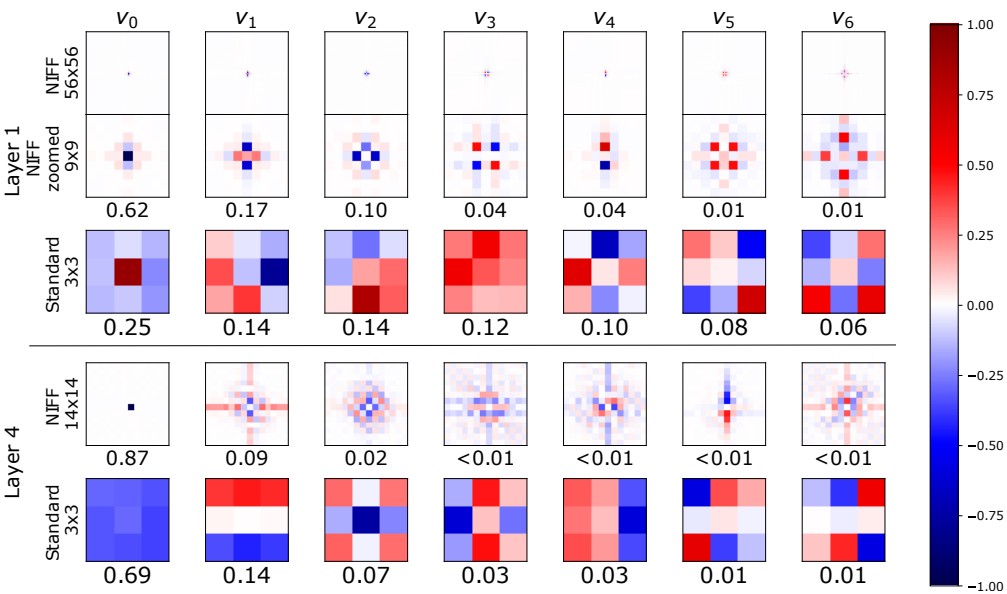

Figure 1: PCA components of learned kernel weights in the first layer of a ResNet50 trained on ImageNet-1k: the 1st row shows the learned NIFF kernels transformed to the spatial domain. Row 2 visualizes a zoomed-in version with explained variance for each component. The 3rd row shows PCA and explained variance for a standard CNN with the standard kernel size $3 \times 3$. The 4th row shows that NIFF actually learns large and highly structured spatial filters for the 4th layer of the same network, while the baseline model is limited to small filters. This PCA analysis is one of the tools we apply to answer the central question of this paper: *How large do CNN kernels really need to be?* It demonstrates that networks which can learn filter kernels as large as their featuremaps, still learn well-localized, small kernels. Yet, these kernels are larger than the typically applied $3 \times 3$ kernels.

the spatial context during encoding in convolutional neural networks (CNNs), leading to improved results for CNNs accordingly, as for example in Guo et al. (2022); Liu et al. (2022a); Ding et al. (2022); Peng et al. (2017). Yet, model parameters and training times scale poorly with the filter size, such that the increased model accuracy often comes at significant costs if no additional model-specific tricks are applied. At the same time, it remains unclear whether there is an optimal filter size and which size of filters would be learned, could the models learn arbitrary sizes.

This paper aims to provide such a study and the corresponding tool that can modularly replace the convolution operation in any CNN architecture allowing for the efficient training of arbitrarily large convolution kernels to be analyzed. However, efficiently training models on standard datasets such as ImageNet Deng et al. (2009) with large filters (potentially as large as the input data) is non-trivial. Not only should the number of parameters remain in a range comparable to the one of the baseline models, but also has the cost of the actual convolution operation to remain manageable. Neural implicit functions, such as previously used in Romero et al. (2022a); Sitzmann et al. (2020), can limit the number of learnable parameters while learning large convolution filters. Yet, their evaluation is limited to low-resolution data because of the poor scalability of the convolution operation itself, i.e. increasing the size of the learned convolution filters directly or implicitly is not a scalable solution. Therefore, we propose to learn filters in the frequency domain via neural implicit functions. This has several advantages: First, the convolution operation can be executed in the Fourier domain, where it scales significantly better with the filter size. Second, due to the implementation of the neural implicit function, the number of learnable model parameters remains similar to that of the baseline model. Third, the learned filters are directly expressed in the Fourier domain i.e. as oriented sine and cosine waves. Thus, highly structured periodic spatial convolution kernels can be learned using small MLPs with only a few parameters. Our proposed **N**eural **I**mplicit **F**requency **F**ilter (NIFF) convolution operation is a plug-and-play module that can replace any conventional convolution in a CNN and allows for an efficient implementation in current frameworks. The resulting neural implicit frequency CNNs are the first models to

achieve results on par with the state-of-the-art on standard high-resolution image classification benchmarks while executing convolutions solely in the frequency domain. Thus, NIFFs allow us to provide an extensive analysis of the practically learned filter sizes while decoupling the filter size from other aspects such as the network width or the number of learnable parameters. Interestingly, our analysis shows that, although the proposed networks could learn very large convolution kernels, the learned filters practically correspond to well-localized and relatively small convolution kernels when transformed from the frequency to the spatial domain. Our contributions can be summarized as follows:

- We present a novel approach which enables CNNs to efficiently learn infinitely large convolutional filters. To do so, we introduce MLP parameterized Neural Implicit Frequency Filters (NIFFs) which learn filter representations directly in the frequency domain and can be plugged into any CNN architecture.

- Empirically, we show that NIFFs facilitate a model performance on par with the baseline without any hyperparameter tuning. Hence, our proposed frequency filters allow, for the first time, to efficiently analyze filter sizes and encoded context sizes - via filters that have potentially an infinite extent in the spatial domain.

- Finally, we analyze the spatial representations of the resulting large filters learned by various CNN architectures and show very interesting results in terms of practically employed spatial extent.

## 2 Related Work

**Large kernel sizes.** In recent years, vision transformers have facilitated a significant improvement in image classification Dosovitskiy et al. (2020); Vaswani et al. (2017) and related tasks. Such models are based on larger image patches and self-attention, i.e. they allow for the entire spatial context to be encoded. Subsequently, Liu et al. showed that smaller receptive fields by shifted window attention with window sizes of $7 \times 7$ to $12 \times 12$ can improve network performance, while follow-up works again increased the window sizes Dong et al. (2022). What remains is the general observation that image-processing models can benefit from encoding larger spatial context, especially in deeper network layers. This holds also for pure convolutional models as shown in Liu et al. (2022b) with $7 \times 7$ sized depth-wise convolutions that can even outperform transformer models. Guo et al. and Peng et al. further increase the receptive fields of the convolution and Ding et al. and Liu et al. achieve improved results with $31 \times 31$ and even $61 \times 61$ sized convolution filters, respectively. To facilitate scaling, Ding et al. use depth-wise convolutions instead of full convolutions and thereby increase the model size by only 10-16%. Liu et al. decompose the $61 \times 61$ convolution into two parallel and rectangular convolutions to reduce the computation load and parameters. Further, Agnihotri et al. (2023) showed that increased spatial context can improve the upsampling in decoder networks. Similarly, Jung & Keuper (2021); Durall et al. (2020) counteract spectral artifacts during image generation using an adversarial loss and regularization combined with large filter sizes, respectively. To allow to further scale even models using depth-wise convolutions, NIFFs directly learn and execute the convolution in the frequency domain. An additional complication of large spatial convolutions are potential spectral artifacts that can be handled e.g. by Hamming windows Tomen & van Gemert (2021).

In sequence modelling, so-called *long convolutions* also enable models to encode larger parts of a sequence at low cost. Several works Poli et al. (2023); Romero et al. (2022b); Gu et al. (2022) make use of the convolution theorem to model global large 1D convolutions. Rao et al. propose Global Filter Networks based on Swin-Transformers. They encode intermediate layers as convolutions in the frequency domain followed by a feed-forward neural network, and thus facilitate efficient convolutions with infinite extent. They achieve very good results on ImageNet-1k. However, they face the computational overhead of learning the filter weights directly, resulting in an increase in the number of parameters. We also compute convolutions in the frequency domain to benefit from the better scalability of the convolution operation. Yet, our NIFF module uses neural implicit functions to preserve the original number of model parameters. The model proposed by Rao et al. is over six times larger than ours. Furthermore, our study focuses on the analysis of the effective kernel size for convolutions employed in CNN backbones and not on Transformer architectures.

**Dynamic and steerable filters.** While our approach focuses on the evaluation of the chosen filter size of the network given that it can learn infinitely large filters in theory, dynamic filtering tries to directly address this challenge by e.g. deformable convolutions Dai et al. (2017). Unlike traditional convolutions where the kernel is fixed and applied uniformly across the input, deformable convolutions allow the filter to adapt its shape and size based on the input data. This is done via a learnable offset that controls the sampling locations of the convolutional operation within the input feature map Further, Pintea et al. (2021) proposed to use flexible kernel sizes at each stage by learning the $\sigma$ values for a combination of Gaussian derivative filters thus inherently controlling the size of the filter. Sosnovik et al. (2019) combine the concepts of steerable filters Cohen & Welling (2016) and group equivariance to achieve scale-equivariant representations, allowing it to effectively handle objects at different scales without the need for explicit scale normalization techniques. Similarly Worrall & Welling (2019) aims to achieve equivariance over scale. They integrate scale-space operators, such as Gaussian blurring and Laplacian of Gaussian operators, into the network layer to be more robust to variations in object size or resolution in images.

**Training CNNs in the Frequency Domain.** Most works implementing convolutions in the frequency domain focus on the efficiency of the time and memory consumption of this operation Ayat et al. (2019); Guan et al. (2019); Mathieu et al. (2013); Pratt et al. (2017); Vasilache et al. (2014); Wang et al. (2016), since the equivalent of convolution in the spatial domain is a point-wise multiplication in the frequency domain. However, most of these approaches still learn the convolution filters in the conventional setting in the spatial domain and transform featuremaps and kernels into the frequency domain to make use of the faster point-wise multiplication at inference time Ayat et al. (2019); Mathieu et al. (2013); Wang et al. (2016). This is mostly due to the fact that one would practically need to learn filters as large as featuremaps when directly learning the frequency representation. Those few but notable works that also propose to learn convolutional filters in CNNs purely in the frequency domain, until now, could only achieve desirable evaluation performances on MNIST and reach low accuracies on more complex datasets like CIFAR or ImageNet Pan et al. (2022); Pratt et al. (2017); Watanabe & Wolf (2021) if they could afford to evaluate on such benchmarks at all. In Table 1 we give a brief overview on some of these approaches, executing the convolution or even more components in the frequency domain. As shown in Table A9 in the appendix, NIFF can also be combined with other modules in the frequency domain.

Further approaches apply model compression by zeroing out small coefficients of the featuremap in the frequency domain Guan et al. (2019), enhance robustness by frequency regularization Lukasik et al. (2023) or enrich conventional convolutions with the frequency perspective to get more global information Chi et al. (2020).

In contrast to all these approaches, we neither aim to be more time or memory-efficient nor do we want to boost performance via additional information. Our aim is to investigate which filter size CNNs practically make use of if they have the opportunity to learn infinitely large kernels. To do so, we propose a model that facilitates learning large filters directly in the frequency domain via neural implicit functions. Thus, we are the first to propose a CNN whose convolutions can be fully learned in the frequency domain while maintaining state-of-the-art performance on common image classification benchmarks.

**Neural Implicit Representations.** Neural implicit representations generally map a point encoded by coordinates via an MLP to a continuous output domain. Thus, an object is represented by a function rather than fixed discrete values, which is the more common case, e.g., discrete grids of pixels for images, discrete samples of amplitudes for audio signals, and voxels, meshes, or point clouds for 3D shapes Chibane et al. (2020); Sitzmann et al. (2020). Continuous representation of the signal, i.e. neural implicit representations, provide a fixed number of parameters and are independent of spatial or frequency resolutions. Moreover, neural implicit representation can also be used to learn to generate high-resolution data Chen et al. (2021), or to learn 3D representations from 2D data Ma et al. (2022) and Ma et al. (2023) introduce hyperconvolutions for biomedical image segmentation. Based on the concept of Hypernetworks Ha et al. (2016), they learn a predefined SIREN Sitzmann et al. (2020) consisting of stacked 1x1 convolutions and periodic activations to predict the spatial kernel weights with predefined size. Similarly, Romero et al. (2022a) introduces continuous kernels by learning spatial kernel weights that have the same size as featuremaps with a Hypernetwork consisting of stacked 1x1 convolutions and then learn to mask and crop the actual filter size. All their

Table 1: Overview of different approaches operating in the frequency domain compared to our NIFF. The approaches in the first two rows (Mathieu et al. (2013); Wang et al. (2016)) focus on the time improvement of the point-wise multiplication compared to standard convolution. The third and fourth row present approaches that operate (almost) fully in the frequency domain (Pratt et al. (2017); Pan et al. (2022)). Tomen & van Gemert (2021) applies a global filter layer via point-wise multiplication in the frequency domain at the stem of a transformer model. In comparison we present NIFF, a plug-and-play operation that can replace any kind of convolution operation to learn and infer fully in the frequency domain. Note that these methods have been proposed and optimized for different purposes and the respective numbers on CIFAR-10, where reported, are not comparable.

| Method | Architecture | Operations in the frequency domain | reported CIFAR-10 acc |
|---|---|---|---|
| Fast FFT [2013] | not reported | Executing of the Convolution | no acc. reported |
| CS-unit [2016] | not reported | Convolution + Downsampling | no acc. reported |
| FCNN [2017] | FCNN | Full Network | $\sim 23\%$ |
| CEMNet [2022] | CEMNet | Full Network except MaxPooling | 59.33%-78.37% |
| GFN [2021] | Swin-Transformer | First Patchifying operation | 98.60% |
| NIFF (ours) | any CNN | Convolution | 90.63%-94.03% |

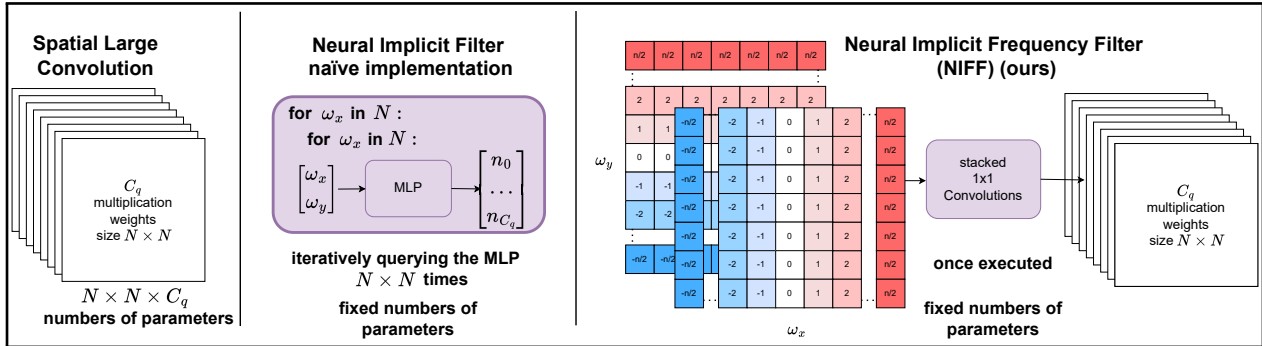

Figure 2: While learning large kernels increases the number of learnable parameters quadratically (here N×N (left)), neural implicit functions use a fixed amount of learnable parameters (middle). When using a simple MLP, the input is a 2D vector containing the $\omega_x$ and $\omega_y$ coordinates of the desired filter position. Our NIFF (right) implements the MLP efficiently using several 1×1 convolutions which start with an input channel size of two, encoding the $\omega_x$ and $\omega_y$ direction. Hence, there is no need to iterate over each coordinate separately. Following, we include some hidden layers and activations to learn and at the last layer, the number of output channels is set to the desired number of element-wise multiplication weights.

operations are applied in the spatial domain. To facilitate the learning of suitable filter structures, they replace the periodic activation of the SIREN by Multiplicative Anisotropic Gabor Networks. In contrast, our NIFFs directly learn the Fourier representation of convolution filters, i.e. a representation in a basis of sine and cosine waves, using an MLP with conventional activations. Thus, we can efficiently execute the convolution in the Fourier domain with the objective of investigating the effective spatial extent of the learned convolution filters.

## 3 Convolutions in the Frequency domain

In this paper, we leverage neural implicit functions in a novel setting: we learn neural implicit representations of the spatial frequencies of large convolution kernels. This allows us to employ the convolution theorem from signal processing Forsyth & Ponce (2003) and conduct convolutions with large kernels efficiently in the frequency domain via point-wise multiplications.

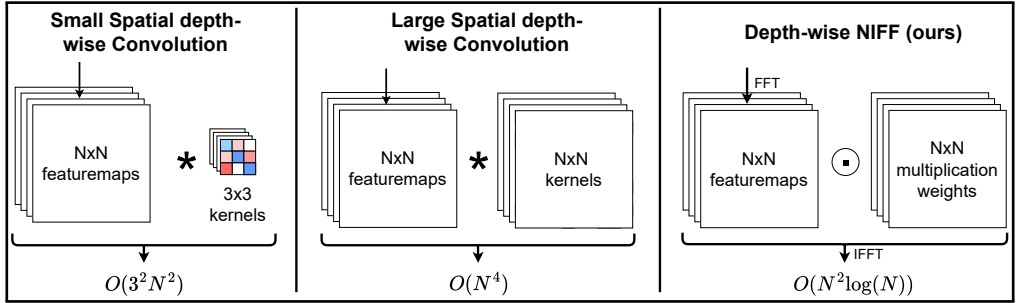

Figure 3: Concept of our NIFF Convolutions and the complexity for each operation, explained with the example of a depth-wise convolution. (Left) The standard depth-wise convolution in which we have as many kernels as we have featuremaps. Each kernel is convolved with a featuremap. (Middle) Large convolution with kernels as large as the featuremaps. (Right) The NIFF convolution which simply applies a pointwise multiplication between the FFT-transformed featuremaps and the learned kernel weight via our NIFF. The newly updated featuremaps are transformed back into the spatial domain via inverse FFT (IFFT).

**Properties of Convolutions in the Frequency Domain.** According to the convolution theorem Forsyth & Ponce (2003), a circular convolution, denoted by $\circledast$, between a signal $g$ and filter $k$ in the spatial domain can be equivalently represented by a point-wise multiplication, denoted by $\odot$, of these two signals in the frequency domain, for example by computing their Fast Fourier Transform (FFT), denoted by the function $\mathcal{F}(.)$ and then, after point-wise multiplication, their inverse FFT $\mathcal{F}^{-1}(.)$:

$$g \circledast k = \mathcal{F}^{-1}(\mathcal{F}(g) \odot \mathcal{F}(k)) \tag{1}$$

While this equivalence has been less relevant for the relatively small convolution kernels employed in traditional CNNs (typically $3 \times 3$ or at most $7 \times 7$), it becomes highly relevant as the filter size increases to a maximum: The convolution operation in the spatial domain is in $O(M^2N^2)$ for a discrete 2D signal $g$ with $N \times N$ samples and filters $k$ of size $M \times M$, i.e. $O(N^4)$ when discrete filters $k$ have the same size as the signal $g$. In contrast, the computation is in $O(N^2\log(N))$ when executed using Fast Fourier Transform (FFT) Cooley & Tukey (1965) and point-wise multiplication according to Equation equation 1. The proof for the FFT according to Cooley & Tukey (1965) is given in the appendix.

Thus, for efficient filter learning, we assume that our input signal, i.e. our input image or featuremap, is given in the frequency domain. There, we can directly learn the element-wise multiplication weights $m$ corresponding to $\mathcal{F}(k)$ in Equation equation 1 and thereby predict infinitely large spatial kernels. These element-wise multiplication weights $m$ act like a circular convolution which can be seen as an *infinite* convolution due to the periodic nature of the frequency domain representation. This means in practice that if we represent a signal, in our case an image or a featuremap, in the frequency domain and transform it back into the spatial domain, we assume that this signal is periodic and thus infinitely large. For many images, such boundary conditions can make sense since the image horizon line is usually horizontally aligned.

Practically, the kernels applied in the frequency domain are band-limited - to the highest spatial frequency that can be represented in the featuremap. However, since higher frequencies can not be encoded in the discrete input signal by definition, this is no practical limitation. In the frequency domain, we can thus efficiently apply convolutions with filters with standard sizes of for example $224 \times 244$ (for ImageNet) or $32 \times 32$ (for CIFAR-10) learned spatial frequencies, solely limited by the resolution of the input featuremap.

## 3.1 Neural Implicit Frequency Filters

Images are typically stored as pixel-wise discrete values. Similarly, the filter weights of CNNs are usually learned and stored as discrete values, i.e. a $3 \times 3$ filter has 9 parameters to be learned and stored, a $7 \times 7$ filter as in ConvNeXt has 49 parameters and a filter as large as the featuremap would require e.g. $224 \times 224$ (50176) parameters for ImageNet-sized network input. In this case, it is not affordable to directly learn these filter weights, neither in the spatial nor in the frequency domain. To address this issue, we propose to parameterize

filters by neural implicit functions instead of learning the kernels directly. This is particularly beneficial since we can directly learn the neural implicit filter representations in the frequency domain. Figure 2 depicts the benefit of neural implicit functions for large kernel sizes.

Thus, formally, we learn a function, $F$ parameterized by $\Phi$ that takes as input the spatial frequency $(\omega_x, \omega_y)$ whose filter value it will predict,

$$F_\Phi : \mathbb{R}^2 \mapsto \mathbb{C}^C, m(\omega_x, \omega_y) := F_\Phi(\omega_x, \omega_y), \tag{2}$$

where $C$ is the number of filter channels and the complex-valued $m(\omega_x, \omega_y)$ in dimension $c$ is the $c$-th filter value in the frequency domain to be multiplied with $\mathcal{F}(g)(\omega_x, \omega_y)$ for featuremap $g$. Specifically, with the implicit function $F_\Phi$, we parameterize the weights with which the featuremaps are multiplied in the frequency domain based on equation 1 by point-wise multiplication. The number of MLP output channels is equivalent to the number of channels $C$ for the convolution, and its hidden dimensions determine the expressivity of each learned filter, which we term *Neural Implicit Frequency Filter (NIFF)*. In practice, the complex and real valued parts of NIFFs are independently parameterized.

**Efficient Parameterization of NIFFs.**   While neural implicit functions allow parameterizing large filters with only a few MLP model weights, their direct implementation would be highly inefficient. Therefore, we resume to a trick that allows efficient training and inference using standard neural network building blocks. Specifically, we arrange the input to the MLP, i.e. the discrete spatial frequencies $(\omega_x, \omega_y)$ for which we need to retrieve filter values, in 2D arrays that formally resemble featuremaps in CNNs but are fixed for all layers. Thus, the MLP takes one input matrix encoding the $x$ coordinates and one encoding the $y$ coordinates as shown in Figure 3. Then, the MLP can be equivalently and efficiently computed using stacked $1 \times 1$ convolutions, where the first $1 \times 1$ convolution has input depth two for the two coordinates, and the output layer $1 \times 1$ convolution has $C$ output dimensions.

### 3.2   Common CNN Building Blocks using NIFF

Well-established models like ResNet He et al. (2016) use full convolutions, while more recent architectures employ depth-wise and $1 \times 1$ convolutions separately Liu et al. (2022b). Our neural implicit frequency filters can be implemented for all these cases. However, operations that include downsampling by a stride of two are kept as original spatial convolution. In the following, we describe how commonly used convolution types are implemented using NIFF.

**Depth-wise Convolution.**   The NIFF module for the depth-wise convolution is as follows: First, we transform the featuremaps into the frequency domain via FFT. Afterwards, the learned filters are applied via element-wise multiplications with the featuremaps. Thereafter, the featuremaps are transformed back into the spatial domain via inverse FFT (if the next step is to be executed in the spatial domain). The entire process is visualized in Figure 3. As discussed above, this process allows to train models with large *circular* convolutions in a scalable way. An ablation on circular versus linear convolutions in the frequency domain is given in the appendix in Table A3.

**Full Convolution.**   To employ NIFF for a full convolution *2DConv* with $C_p$ input channels and $C_q$ output channels the convolved featuremaps with the kernel weights need to be summed according to

$$2\text{DConv}(g_{C_p}, k_{C_p, C_q}) = \sum_c^{C_p} g_c \circledast k_{c, C_q} = g_{C_q}. \tag{3}$$

Conveniently, a summation in the spatial domain is equivalent to a summation in the frequency domain and can be performed right away.

$$g + k = \mathcal{F}^{-1}(\mathcal{F}(g) + \mathcal{F}(k)) \tag{4}$$

The full convolution in the frequency domain can be implemented by first predicting the frequency representation of $k_{C_p, C_q}$ directly using NIFF, i.e. for each output channel. Then, all input channels are element-wise

multiplied with the filter weights and summed up in 2DConv$_{NIFF}$:

$$\sum_c^{C_p} g_c \circledast k_{c,C_q} = \mathcal{F}^{-1} \left( \sum_c^{C_p} \mathcal{F}(g_c) \odot \mathcal{F}(k_{c,C_q}) \right) \tag{5}$$

Yas the NIFF needs to output $C_p \times C_q$ instead of only $C_q$ at its last layer leading to a significant increase of learnable parameters for our NIFF. Thus, for efficiency reasons, we decompose the full convolution into a depth-wise convolution followed by a $1 \times 1$ convolution in practice. The transformation into the frequency domain is applied before the depth-wise convolution, where the backward transformation into the spatial domain is applied after the $1 \times 1$ convolution. While not equivalent, the amount of learnable parameters decreases significantly with this approach, and the resulting models show similar or better performance in practice. An ablation on full convolutions versus depth-wise separable convolutions is provided in Table 2 and in the appendix in Table A1.

$1 \times 1$ **Convolution.** To perform a $1 \times 1$ convolution, we transform the input into the frequency domain via FFT. Afterwards, we apply a linear layer with channel input neurons and desired output dimension output neurons on the channel dimension. Finally, we transform back into the spatial domain via inverse FFT. While spatial $1 \times 1$ convolutions only combine spatial information in one location, our $1 \times 1$ in the frequency space is able to combine and learn important information globally.

Other operations such as downsampling, normalization, and non-linear activation are applied in the spatial domain so that the resulting models are as close as possible to their baselines while enabling infinite-sized convolutions.

## 4  NIFF Model Evaluation

We evaluate a variety of image classification CNNs with our proposed NIFF. Overall, we achieve accuracies on par with the respective baselines. For high-resolution data, our NIFF CNNs perform slightly better than the baseline models, while the large kernels are apparently less beneficial for low-resolution data, especially in deeper layers where the featuremap sizes are very small. Further, we evaluate the performance of NIFF when combined with other methods applied in the frequency domain in the appendix (Appendix E).

**Results on Image Classification.** For high-resolution datasets like ImageNet-1k or ImageNet-100, NIFF CNNs are able to achieve results on par with the baseline models. Table 2 and Table A1 in the appendix report these results. For models which originally employ full 2D convolutions, we also ablate on the effect of replacing these with depth-wise separated convolutions in spatial domain with the original filter size and for the smaller ResNet-18 and ResNet-50 models, we report results on NIFF for full 2D convolution. In line with the finding e.g. in Liu et al. (2022b), the differences are rather small while the amount of parameters decreases significantly as depth-wise separable convolutions are applied. According to our results, this is equally true for both, convolutions executed in the spatial and in the frequency domain. For comparability, we simply used the baseline hyperparameters reported for each of these models in the respective papers, i.e. we achieve results on par with the respective baselines without hyperparameter optimization. For different ResNet architectures, we even achieve improvements on both high-resolution datasets.

The complete results on CIFAR-10 are reported in Table A2 in the Appendix. Here, we observe a slight drop in performance, which may be due to the low spatial resolution of images and featuremaps. Particularly in deeper layers, the potential benefit from large convolutions is therefore limited. Further, we want to emphasize that NIFF is not conceived to improve over baseline methods (we are of course glad to observe this tendency for high-resolution data) but to facilitate the evaluation of effective kernel sizes in CNNs.

## 5  How large do spatial kernels really need to be?

After empirically validating our proposed NIFFs, we now quantitatively analyze how large the spatial kernels really tend to be. Hence, we transform the learned filters into the spatial domain and plot the relative density

Table 2: Evaluation of top 1 and 5 test accuracy on ImageNet-1k and ImageNet-100 for different network architectures. We used the standard training parameter for each architecture and the advanced data augmentation from Liu et al. (2022b) for all architectures. Additionally, we add the numbers reported by Pytorch for the *timm* baseline models on ImageNet-1k for each network. More ablations on ImageNet-100 regarding more architectures and evaluating linear convolutions are given in the appendix Table A1 and Table A3.

| Model | ImageNet-1k | | | ImageNet-100 | | |
|---|---|---|---|---|---|---|
| | # Params | Acc@1 | Acc@5 | # Params | Acc@1 | Acc@5 |
| ResNet-18 He et al. (2016) (Pytorch) | 11.689.512 | 69.76 | 89.08 | 11.227.812 | - | - |
| ResNet-18 He et al. (2016) | 11.689.512 | **72.38** | **90.70** | 11.227.812 | **87.52** | **97.50** |
| ResNet-18 NIFF 2D conv (ours) | 21.127.320 | 71.00 | 89.95 | 20.665.620 | 86.42 | 97.08 |
| ResNet-18 separated conv | 3.327.400 | 69.12 | 88.84 | 2.865.700 | 86.52 | 97.20 |
| ResNet-18 NIFF (ours) | 3.660.360 | 70.75 | 90.01 | 3.198.660 | 86.52 | 97.14 |
| ResNet-50 He et al. (2016) (Pytorch) | 25.557.032 | 76.13 | 92.86 | 23.712.932 | - | - |
| ResNet-50 He et al. (2016) | 25.557.032 | 79.13 | 94.43 | 23.712.932 | 89.88 | 98.22 |
| ResNet-50 NIFF 2D conv (ours) | 33.778.328 | 78.65 | 94.25 | 31.934.228 | **90.08** | 98.06 |
| ResNet-50 separated conv | 18.275.688 | 77.76 | 93.72 | 16.431.588 | 89.78 | 98.18 |
| ResNet-50 NIFF (ours) | 18.605.000 | **79.65** | **94.80** | 16.760.900 | 89.98 | **98.44** |
| ResNet-101 He et al. (2016) (Pytorch) | 44.549.160 | 77.37 | 93.55 | 42705.060 | - | - |
| ResNet-101 He et al. (2016) | 44.549.160 | **80.63** | 95.11 | 42.705.060 | **90.54** | 98.14 |
| ResNet-101 separated conv | 28.394.088 | 79.53 | 94.64 | 26.549.988 | 90.20 | 98.36 |
| ResNet-101 NIFF (ours) | 29.187.432 | 80.26 | **95.23** | 27.343.332 | **90.54** | **98.38** |
| DenseNet-121 Huang et al. (2017) (Pytorch) | 7.978.856 | **76.65** | 92.17 | 7.978.856 | - | - |
| DenseNet-121 Huang et al. (2017) | 7.978.856 | 75.11 | **92.50** | 7.056.356 | 90.06 | 98.20 |
| DenseNet-121 NIFF 2D conv (ours) | 10.119.752 | 73.96 | 91.82 | 9.197.252 | 89.66 | 98.32 |
| DenseNet-121 separated conv | 6.145.128 | 69.80 | 89.02 | 5.222.628 | 89.94 | 98.08 |
| DenseNet-121 NIFF (ours) | 6.159.512 | 74.58 | 92.33 | 5.237.012 | **90.24** | 98.18 |
| ConvNeXt-tiny Liu et al. (2022b) (PyTorch) | 28.589.128 | **82.52** | **96.15** | 28.589.128 | - | - |
| ConvNeXt-tiny Liu et al. (2022b) | 28.589.128 | - | - | 27.897.028 | 91.70 | 98.32 |
| ConvNeXt-tiny NIFF (ours) | 28.890.664 | 81.83 | 95.80 | 28.231.684 | **92.00** | **98.42** |
| MobileNet-v2 Sandler et al. (2018) (Pytorch) | 3.504.872 | **71.88** | 90.29 | 3.504.872 | - | - |
| MobileNet-v2 Sandler et al. (2018) | 3.504.872 | 70.03 | 89.54 | 2.351.972 | 84.06 | 96.52 |
| MobileNet-v2 NIFF (ours) | 3.512.560 | 71.53 | **90.50** | 2.359.660 | **85.46** | **96.70** |

of each spatial kernel $k$, i.e. the ratio of the kernel mass that is contained within centered, smaller sized, squared kernels. The kernel has the same width and height as the featuremap (FM) it is convolved to.

$$\text{kernel mass ratio(width, height)} = \frac{\sum_{w=1}^{\text{width}} \sum_{h=1}^{\text{height}} k\left(c - \lfloor \frac{\text{width}}{2} \rfloor + w, c - \lfloor \frac{\text{height}}{2} \rfloor + h\right)}{\sum_{w=1}^{\text{FMwidth}} \sum_{h=1}^{\text{FMheight}} (k(w,h))}, \tag{6}$$

where $c$ is the center of the full kernel $k$ in the spatial domain. In Figure 4 we report the kernel mass ratio for different networks trained on ImageNet-1k. We observe that all networks mostly contain well-localized kernels that are significantly smaller than possible. Yet, the first layer in DenseNet-121 and the second and third layer in ResNet-50 also contain larger kernels that make use of the possible kernel size up to $56 \times 56$ and $28 \times 28$ respectively. For MobileNet-v2, the spatial kernels are predominantly well-localized and small. However, at the maximal resolution of $112 \times 112$ some kernels are quite large, at least $56 \times 56$ ( 15%) indicating that MobileNet-v2 which uses small $3 \times 3$ kernels could benefit from larger kernels. Similar results

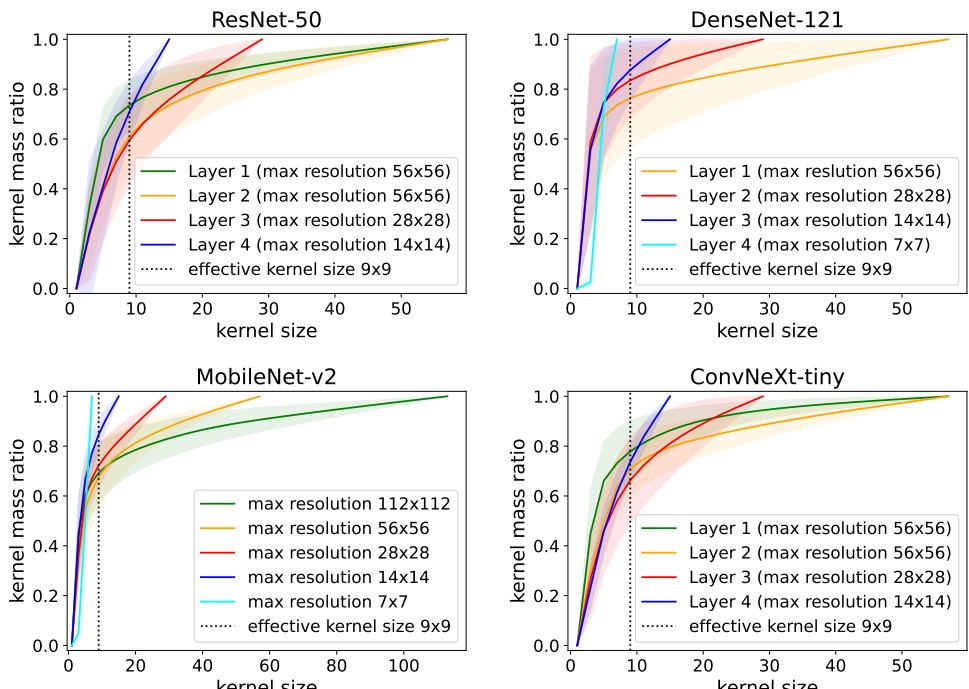

Figure 4: Effective kernel size evaluation on ImageNet-1k. We plot the average ratio of the entire kernel mass contained within the limited spatial kernel size, where the x-axis denotes the width and height of the squared kernels. For ResNet-18, ResNet-50 and ConvNeXt-tiny each layer encodes one resolution. Thus, these network's layers could be summarised (Layer 1 encoding $56 \times 56$, Layer 2 $28 \times 28$, Layer 3 $14 \times 14$ and Layer 4 $7 \times 7$). However, for MobileNet-v2 the resolution is downsampled within a layer.

on ImageNet-100 are reported in the Appendix (Figure A2). These findings are in line with Romero et al. (2022a), who investigate the spatial kernel size of CNNs in the spatial domain.

In Figure 5, we evaluate the spatial kernel mass ratio of different networks trained on CIFAR-10. For all models, we see a clear trend. They learn in all layers well-localized, small kernels. Similar to the ResNet trained on ImageNet-1k, the learned kernels barely exceed the size of $5 \times 5$. In contrast, some of the learned spatial kernels by MobileNet-v2 and ConvNeXt-tiny use the full potential for each kernel size, respectively.

# 6 Filter Analysis

In this section, we visualize the spatial kernels learned by our NIFF. We do so by transforming the learned multiplication weights via inverse FFT into the spatial domain. Afterwards, we apply Principle Component Analysis (PCA) per layer to evaluate the predominant structure of the learned spatial kernels. For each layer, we separately plot the six most important eigenvectors and zoom in to visualize the $9 \times 9$ center (see Appendix Fig. A8 for the full visualization without center cropping). The original spatial kernels are shown in the Appendix (Fig. A22 and A23).

Our results indicate that the networks, although they could learn infinitely large kernels, learn well-localized, quite small ($3 \times 3$ up to $9 \times 9$) kernels especially in early layers. The spatial kernel size is different for low- and high-resolution data. Figure 6 visualizes the eigenvectors of the spatial kernel weights for each layer in our NIFF CNN for ResNet-50 trained on ImageNet-1k. All layers dominantly learn filters with well-localized, small kernel sizes. Especially in the later layers, most variance is explained by simple $1 \times 1$ kernels, whereas the first-layer kernels have the most variance explained by simple $5 \times 5$ or $7 \times 7$ kernels. Hence, the networks learn larger kernels than the standard $3 \times 3$ kernels used in a ResNet. However, the full potential of possible sizes of $56 \times 56$ and more is not used. Similar results are shown in Figure 1 as well as for more architectures (Figure A5, A6 and A7) and ImageNet-100 (Figure A9, A10, A11 and A12) in the Appendix with consistent

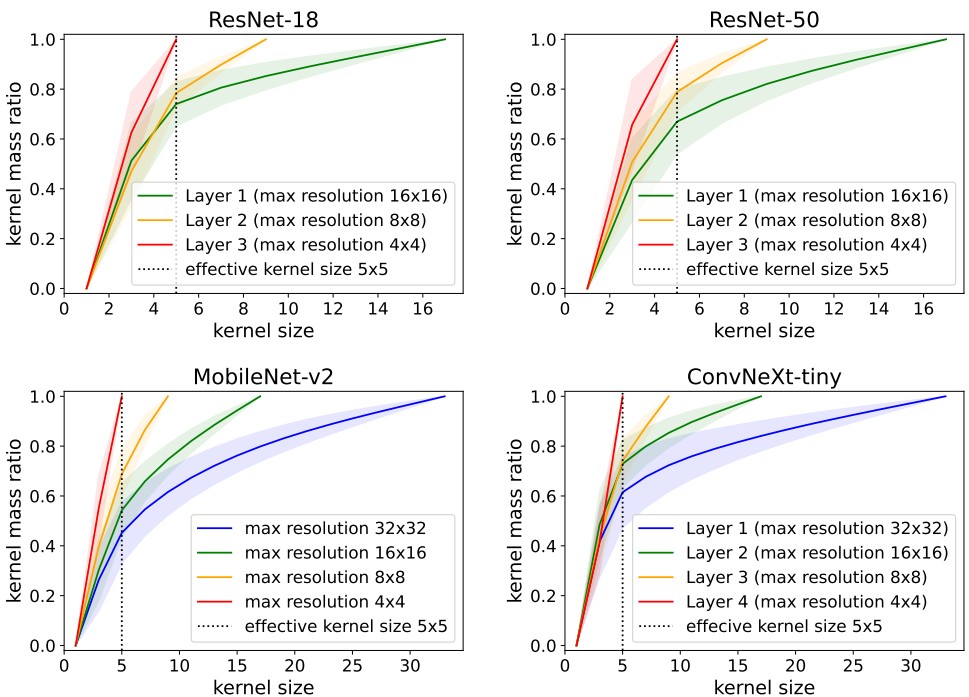

Figure 5: Effective kernel size evaluation on CIFAR-10. We plot the average ratio of the entire kernel mass contained within the limited spatial kernel size, where the x-axis denotes the width and height of the squared kernels. For ResNet models, each layer encodes one resolution. Thus, the layers could be summarised (Layer 1 encoding $16 \times 16$, Layer 2 $8 \times 8$ and Layer 3 $4 \times 4$). For ConvNeXt-tiny the first layer started with $32 \times 32$. However, for MobileNet-v2 the resolution is downsampled within a layer.

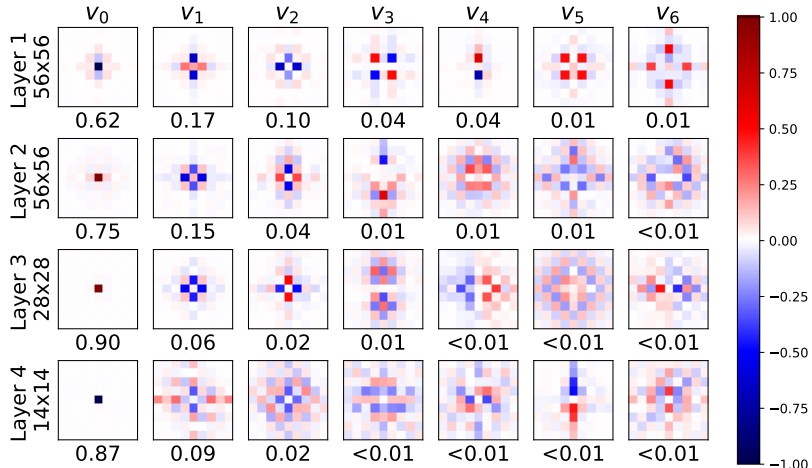

Figure 6: PCA basis and explained variance for each basis vector (below) of all spatial filters for each layer of a ResNet-50 trained on ImageNet-1k zoomed to $9 \times 9$. On the left, the maximal filter size for the corresponding layer is given. We can see that most filters only use a well-localized, small kernel size although they could use a much bigger kernel.

results; all networks learn well-localized, small kernels, even though they could learn much larger ones. Some exceptions can be observed for the 4th layer, where the eigenvector with the second highest explained variance still uses a larger extent of the kernel. In combination with the downsampling applied prior to the

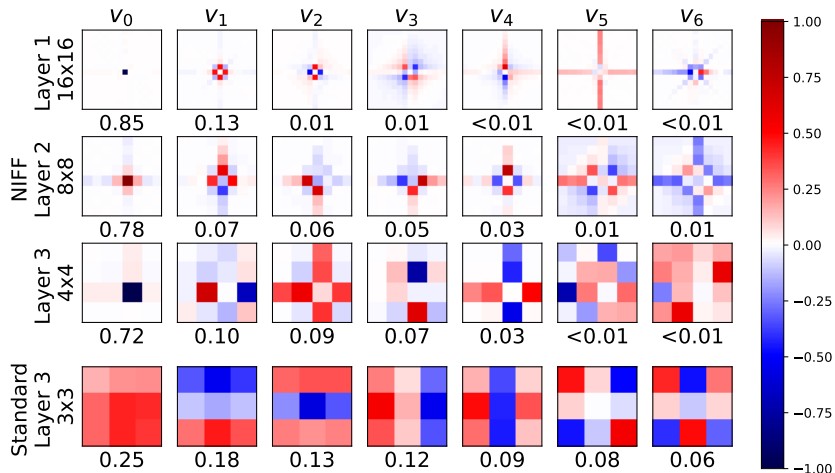

Figure 7: PCA basis and explained variance for each basis vector (below) of all spatial filters learned by NIFF for each layer as well as the learned filters for the third layer of a standard ResNet-18 trained on CIFAR-10. On the left, the layer and its filter size are given. Most filters only use a well-localized, small kernel size although they could use a much bigger kernel.

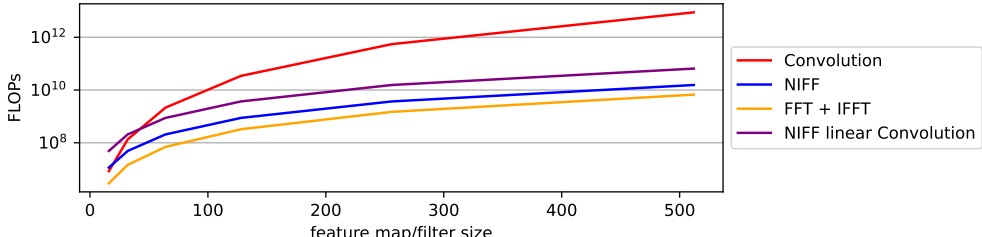

Figure 8: FLOPs in Log-scale for computing a simple FFT and IFFT, a standard depth-wise convolution and our NIFF (including FFT and IFFT) and a linear convolution executed with out NIFF for convolutions with kernels as big as the featuremaps for the example of 64 channels.

fourth layer, this indicates that the network tries to keep up a large spatial context. The same effect can be observed for networks trained on CIFAR-10. Figure 7 shows the PCAs of the spatial kernel weights for each layer in our NIFF CNN for a ResNet18 trained on CIFAR-10. There the dominant basis vectors for the first layer learned by our NIFF kernels do not exceed a size of $3 \times 3$ in the spatial domain. However, in the second layer, a few kernels (19%) are larger than $3 \times 3$ mostly $7 \times 7$. In the last layer, the kernels again do not exceed the standard size of $3 \times 3$. Similar results on further networks are presented in the Appendix (Figure A13).

**Compute Costs.** Since our NIFF implementation is conceived for analysis purposes, our models are not optimized for runtime. In particular, we compute repeated FFTs in *Pytorch* to allow the computation of the remaining network components in the spatial domain, so that the models are equivalent to large kernel models computed in the spatial domain. Yet, Figure 8 demonstrates that with large kernel sizes our NIFF approach with repeated FFTs is much more efficient in terms of FLOPs compared to the spatial convolution. Further runtime evaluations are reported in the Appendix (Tables A5 and A6).

## 7 Discussion and Conclusion

With the help of the proposed NIFF, we can analyze the effectively learned kernel size of state-of-the-art CNNs in the spatial domain. We could observe that the full potential for much larger receptive fields is not used in most models. Especially for low-resolution datasets like CIFAR-10, the network refuses to learn much larger kernels than $3 \times 3$. However, in high-resolution datasets such as ImageNet, the models indeed use

larger kernel sizes than $3 \times 3$ but still the majority of filters is not much larger than $9 \times 9$. Therefore, networks with small kernel sizes such as ResNet He et al. (2016) can benefit from convolution kernels through NIFFs, while networks such as ConvNeXt Liu et al. (2022b) are already close to optimal for ImageNet in terms of used kernel size. However, we also find that there is no globally optimal filter size, as the networks also learn a small amount of large kernels. Thus, we might need to rethink the current state-of-the-art approach of using fixed kernel sizes overall.

Concluding, we propose NIFF CNNs, a tool with which we can learn convolution filters in the frequency domain that translate into infinitely large kernels in the spatial domain. NIFF can efficiently replace common spatial convolutions with element-wise multiplication in the frequency domain. We analyze the resulting kernels in the spatial domain and observe that they are well localized and mostly quite small $(9 \times 9)$. On high-resolution datasets, NIFF can perform on-par with or better than spatial convolutions and can leverage the benefit of encoding large spatial context.

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

# As large as it gets – Studying Infinitely Large Convolutions via Neural Implicit Frequency Filter

## Supplementary Material

In the following, we provide additional information and details that accompany the main paper:

## A   Kernel Mass Evaluation

In this section, we evaluate the kernel mass ratio for more ResNet models trained on ImageNet-1k (Figure A1) and different network architectures trained on ImageNet-100 (Figure A2). The networks show similar behavior already observed in the main paper, all models predominately learn small, well-localized kernels regardless of the potential to learn much larger kernels. However, the smaller ResNet-18 model learns larger kernels than the ResNet-50 or ResNet-101 in the second layer. For ImageNet-100, MobileNet-v2 does not learn as large kernels as observed for ImageNet-1k. Further, ResNet-50 trained on ImageNet-100 seems to learn larger kernels in the second layers compared to the ResNet-50 trained on ImageNet-1k (Figure 4).

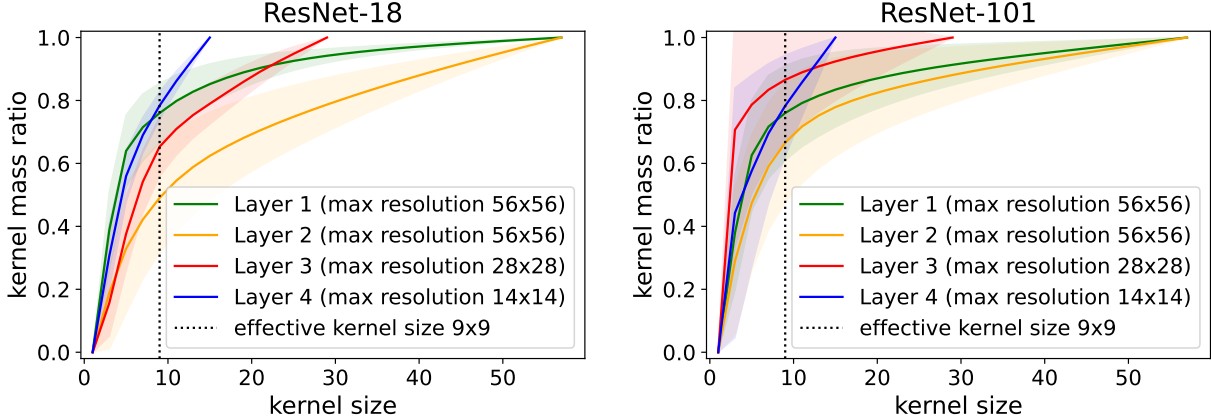

Figure A1: Effective kernel size evaluation on ImageNet-1k for further ResNet models. We plot the average ratio of the entire kernel mass contained within the limited spatial kernel size, where the x-axis denotes the width and height of the squared kernels. The layers are summarised as follows: Layer 1 encoding $56 \times 56$, Layer 2 $28 \times 28$, Layer 3 $14 \times 14$ and Layer 4 $7 \times 7$.

**Non-square kernels**   Following Romero et al. (2022a), we analyse not only square shape kernels, which are typically applied in CNNs but also rectangular shape kernels. Thus, we fit a 2D Gaussian to kernels learned using our NIFF and compare the variance given by $\sigma_x$ and $\sigma_y$ in the x- and y- direction. In detail, we build the ratio between $\sigma_x$ and $\sigma_y$ of the Gaussian. To aggregate over all kernels within one layer, we plot the mean and standard deviation of these ratios in Figure A3 and Figure A4. The mean over all kernels within a layer is near to a ratio of one indicating that most kernels exhibit square-shapes. For some layers (the last layer for ResNet-50 and ResNet-101 on ImageNet-1k and ResNet-18 and ResNet-50 on ImageNet-100) the variance is quite high, indicating that $\sigma_x$ and $\sigma_y$ differ and non-square, rectangle kernels are learned. Not that the learned kernels by Romero et al. (2022a) are parameterized by a Siren Sitzmann et al. (2020) leading to more wave-like, smooth kernels. In contrast, we learn the kernels in the frequency domain which could be wave-like, but are mostly not wave-like as shown in Figure A22 and Figure A23. Therefore, the measured standard deviations in $x$ and $y$ direction should not be understood as a kernel mask as argued in Romero et al. (2022a). They merely indicate the rough spatial distribution of filter weights.

## B  Filter Visualization

### B.1  Principle Component Analysis

The Principle Component Analysis, short PCA, is typically a dimensionality reduction method. The goal of PCA is to maintain most information of a dataset while transforming it into a smaller one. The first principle component explains the most variance of the data, thus representing the majority. The second principle component explains the next highest variance while being orthogonal to the first. For more details on PCA, we refer to Dunteman (1989). For our analysis, we use the PCA of the learned kernels to visualize the predominate structure. Hence, we use the dimensionality reduction property of the PCA to simplify the visualization of the kernels. We also provide images of the original kernels in Figure A22 and A23.

### B.2  Spatial Kernels

In this section, we show the PCA evaluation of the learned spatial kernels by the NIFFs for additional architectures and datasets (ImageNet-100). The results are similar to the ones in the paper (Figure 6, Figure 1 and Figure 7. The learned filters in the spatial domain are well-localized and relatively small compared to the size they could actually learn. This holds true for different architectures on ImageNet-1k (Figure A5, A6 and A7) as well as for ImageNet-100 (Figure A9, A10 and A12).

Further, we show a grid of the actually learned filter in the spatial domain in Figure A22 and A23.

The learned spatial filters on CIFAR-10 are shown in Figure A13. Similarly, to the results shown in Figure 7. The network learns well-localized, small filters n the spatial domain. Yet, for the feature map size of $8 \times 8$ in Layer 2 the network uses significantly more than the standard kernel size of $3 \times 3$.

### B.3  NIFF multiplication weights

Moreover, we analyze the learned element-wise multiplication weights for the real and imaginary parts of different models trained on ImageNet-1k in the frequency domain. Figures A14, A15, A16 and A17 show the PCA per layer for the learned element-wise multiplication weights for ResNet-50, DenseNet-121, ConvNeXt-tiny and MobileNet-V2 respectively. For ResNet-50 and ConvNext-tiny, it seems as if the networks focus in the first layer on the middle-frequency spectra and in the later layers more on the high-frequency spectra. The multiplication weights learned for MobileNet-V2 (Figure A16) focus in the first layer on low-frequency information in the second layer on high-frequency information and in the third layer again on low-frequency information. The DenseNet-121 (Figure A17) learns high-frequency information prior in the first two layers and low-frequency information predominately in the later, third layer. Hence, a general claim for different models and their learned multiplication weights in the frequency domain can not be derived from our empirical analysis. Still for all networks, the imaginary part seems to be less important for these networks and thus the learned structures are less complex. This might be owed to the fact that with increased sparsity through the activation function in the network, the network favours cosine structures (structures with a peak in the center) over sine structures.

## C  Performance Evaluation

**ImageNet-100** We report the accuracy our NIFF CNNs could achieve on ImageNet-100 and the number of learnable parameters in Table A1. The trend is similar to ImageNet-1k, the larger models benefit from NIFF while the lightweight models do not so much. In addition to the models considered in the main paper, we additionally evaluated MobileNet-v3 Howard et al. (2019). We observe that both the baseline model and the NIFF version have comparably low accuracy. For MobileNets, the training pipeline is usually highly optimized for best performance. The data augmentation scheme from Liu et al. (2022b), that we employ for all trainings to achieve comparable results, does not seem to have a beneficial effect here, neither on ImageNet-1k (for Sandler et al. (2018), Table 2) nor on ImageNet-100 (for Sandler et al. (2018); Howard et al. (2019), Table A1).

Table A1: Performance evaluation of top 1 and 5 test accuracy on ImageNet-100 for different network architectures. We used the standard training parameter for each network architecture and stayed consistent with these for each architecture respectively. For the bigger networks like ResNet and DenseNet, which include 2D convolutions, we split into depth-wise and 1×1 convolution as described in Section 3.2 to reduce the number of parameters, for faster training. For all models, our NIFF CNNs perform slightly better, even with reduced number of parameters.

| Name | # Parameters | Acc@1 | Acc@5 |
|------|-------------|-------|-------|
| ConvNeXt-tiny Liu et al. (2022b) | 27.897.028 | 91.70 | 98.32 |
| NIFF (ours) | 28.231.684 | **92.00** | **98.42** |
| ResNet-18 He et al. (2016) | 11.227.812 | **87.52** | **97.50** |
| NIFF 2D conv (ours) | 20.665.620 | 86.42 | 97.08 |
| ResNet-18 separated conv | 2.865.700 | 86.52 | 97.20 |
| NIFF (ours) | 3.198.660 | 86.52 | 97.14 |
| ResNet-50 He et al. (2016) | 23.712.932 | 89.88 | 98.22 |
| NIFF 2D conv (ours) | 31.934.228 | **90.08** | 98.06 |
| ResNet-50 separated conv | 16.431.588 | 89.78 | 98.18 |
| NIFF (ours) | 16.760.900 | 89.98 | **98.44** |
| ResNet-101 He et al. (2016) | 42.705.060 | **90.54** | 98.14 |
| NIFF 2D conv (ours) | 60.954.180 | 89.94 | 98.14 |
| ResNet-101 separated conv | 26.549.988 | 90.20 | 98.36 |
| NIFF (ours) | 27.343.332 | **90.54** | **98.38** |
| DenseNet-121 Huang et al. (2017) | 7.056.356 | 90.06 | 98.20 |
| NIFF 2D conv (ours) | 9.197.252 | 89.66 | **98.32** |
| DesNet-121 separated conv | 5.222.628 | 89.94 | 98.08 |
| NIFF (ours) | 5.237.012 | **90.24** | 98.18 |
| MobileNet-v2 Sandler et al. (2018) | 2.351.972 | 84.06 | 96.52 |
| NIFF (ours) | 2.359.660 | **85.46** | **96.70** |
| MobileNet-v3 Howard et al. (2019) | 1.620.356 | 79.84 | 94.76 |
| NIFF (ours) | 1.617.748 | 78.90 | 95.40 |

**CIFAR-10** Although NIFF CNNs can perform on par with the respective baseline on high-resolution datasets, their performance is limited on low-resolution dataset. Table A2 shows the results on CIFAR-10 with different architectures. Unfortunately, our NIFF CNNs lose around 1 to 3 % points compared to the baseline models. This can be addressed to our previous observation: The networks trained on CIFAR-10 do only use a small amount of the potential kernel size NIFF provides being as big as the kernels of the baseline model ($3 \times 3$).

## D Circular vs Linear Convolution

Our NIFF as proposed above performs a circular convolution, which allows us to directly apply the convolution theorem and execute it as multiplication in the frequency domain. However, standard convolutions in CNNs are finite linear convolutions. A circular convolution can mimic a linear convolution when zero-padding a signal with length $M$ and a kernel with length $K$ to length $L \leq M + K - 1$ Winograd (1978). Thus, to ablate on circular versus finite linear convolutions, the input featuremaps with size $N \times N$ are

Table A2: Performance evaluation of different networks trained on CIFAR-10 and the number of learnable hyperparameters for each network. To be comparable between all models and architecture changes we used the same training schedule for all of them. One can see that NIFF CNNs perform slightly better with a ConvNeXt Liu et al. (2022b) backbone. However, for other architectures, it performs slightly worse.

| Method | # Parameters | Top 1 Acc |
|---|---|---|
| ConvNeXt-tiny Liu et al. (2022b) | 6.376.466 | 90.37 |
| NIFF (ours) | 6.305.746 | **91.48** |
| ResNet-18 He et al. (2016) | 11.173.962 | **92.74** |
| NIFF 2D conv (ours) | 20.613.546 | 92.66 |
| ResNet-18 separated conv | 2.810.341 | 90.18 |
| NIFF (ours) | 1.932.432 | 90.63 |
| ResNet-50 He et al. (2016) | 23.520.842 | **93.75** |
| NIFF 2D conv (ours) | 31.743.914 | 93.39 |
| ResNet-50 separated conv | 16.237.989 | 92.13 |
| NIFF (ours) | 15.491.600 | 93.11 |
| DenseNet-121 Huang et al. (2017) | 6.956.426 | **93.93** |
| NIFF 2D conv (ours) | 9.099.098 | 92.47 |
| DenseNet-121 separated conv | 5.121.189 | 92.00 |
| NIFF (ours) | 5.555.856 | 92.49 |
| MobileNet-v2 Sandler et al. (2018) | 2.236.682 | **94.51** |
| NIFF (ours) | 2.593.760 | 94.03 |
| MobileNet-v3 Howard et al. (2019) | 1.528.106 | 86.28 |
| NIFF (ours) | 1.526.466 | **86.60** |

zero-padded to $2N \times 2N$. For both, the linear and the circular case, NIFF learns filters with the original size $N \times N$ of the featuremap. To mimic linear filters, the learned filters by our NIFF are transformed into the spatial domain and zero-padded similarly to the input featuremaps to $2N \times 2N$. Afterwards, they are transformed back into the frequency domain and the point-wise multiplication is executed. Note that this is not efficient and just serves the academic purpose of verifying whether any accuracy is lost when replacing linear convolutions by circular ones in our approach. However, the resulting networks experience a performance drop compared to the baseline and our NIFF as shown in Table A3 and Table A4. We hypothesize that this drop in performance results from the enforcement of really large kernels. The additional padding mimics linear finite convolutions that are as big as the featuremaps. Related work has shown that larger context can improve model performance Ding et al. (2022); Liu et al. (2022a). Still, there is a limit to which extent this holds as with large kernels artifacts may arise Tomen & van Gemert (2021). Thus, enforcing kernels as large as the featuremaps seems to be not beneficial as shown by our quantitative results. Another explanation for the drop in accuracy could be the introduction of sinc interpolation artifacts into the padded and transformed featuremaps and kernels. The padding is formally a point-wise multiplication with a box-function in the spatial domain. Thus, sinc-interpolation artifacts in the frequency domain can arise. Figure A20 and Figure A21 show that the learned spatial kernels are larger than the learned kernel when we apply a circular convolution with our NIFF. While the first and second layers still learn relatively small filters compared to the actual size they could learn, the third and fourth layers make use of the larger kernels. Since these results come with a significant drop in accuracy, we should however be careful when interpreting these results.

Table A3: Evaluation of top 1 and top 5 accuracies on ImageNet-100 for networks learning filters with our NIFF but afterwards those are padded in the spatial domain and transformed back into the frequency domain to mimic linear convolutions. All ResNet architectures and DenseNet-121 were trained with separated depth-wise and 1x1 convolutions due to efficiency reasons. All models using finite linear convolutions perform significantly worse than the baseline and our NIFF which applies circular convolutions. This observation is consistent with low-resolution data like CIFAR-10.

| Name | Acc@1 | Acc@5 |
|---|---|---|
| ConvNeXt-tiny Liu et al. (2022b) | 91.70 | 98.32 |
| NIFF (ours) | **92.00** | **98.42** |
| NIFF linear | 83.00 | 95.36 |
| ResNet-18 separated conv | 86.52 | 97.20 |
| NIFF (ours) | 86.52 | 97.14 |
| NIFF linear | 81.90 | 95.42 |
| ResNet-50 separated conv | 89.78 | 98.18 |
| NIFF (ours) | 89.98 | 98.44 |
| NIFF linear | 86.76 | 97.16 |
| ResNet-101 separated conv | 90.20 | 98.36 |
| NIFF (ours) | 90.54 | **98.38** |
| NIFF linear | 86.70 | 97.06 |
| DesNet-121 separated conv | 89.94 | 98.08 |
| NIFF (ours) | **90.24** | 98.18 |
| NIFF linear | 81.40 | 95.52 |
| MobileNet-v2 Sandler et al. (2018) | 84.06 | 96.52 |
| NIFF (ours) | **85.46** | **96.70** |
| NIFF linear | 73.90 | 93.16 |

## E  Runtime

As discussed in the computing costs section of the main paper, our approach is slower than the current implementation with spatial convolutions due to the repetitive use of FFT and IFFT. However, when comparing the number of FLOPs needed to compute convolutions with kernel sizes as big as the featuremaps to our NIFF approach, NIFF requires significantly fewer FLOPs, especially with increased featuremap size. Figure 8 shows that most of the FLOPs for our NIFF result from the additional FFT and IFFT operation. Still, we require much fewer FLOPs than large spatial convolutions.

Moreover, we evaluate the runtime per epoch for each model on CIFAR-10 (Table A5) and ImageNet-100 (Table A6) and compare it to the standard spatial $3\times3$ convolution, which has a much smaller spatial context than our NIFF as well as spatial convolutions which are as large as the featuremaps. This would be comparable to our NIFF. Obviously, small spatial kernels ($3 \times 3$) are much faster than larger kernels like NIFF or large spatial kernels. However, NIFF is much faster than the large spatial kernels during training. Especially on high-resolution datasets like ImageNet-100 our NIFF is over four times faster on ResNet-50 and over three times faster on ConvNeXt-tiny compared to the large convolution in the spatial domain.

In general, we want to emphasize that our NIFF models still learn infinite large kernels while all kernels in the spatial domain are limited to the set kernel size. If one would like to learn a 2D convolution in the spatial domain with an image $g$ of size $N \times N$ and filters with the same size $N \times N$ this would be in $O(N^4)$ whereas using FFT and pointwise multiplication (Equation 1) would result in $O(N^2\log(N))$.

Table A4: Evaluation of top 1 on CIFAR-10 for networks learning filters with our NIFF but afterwards those are padded in the spatial domain and transformed back into the frequency domain to mimic linear, non-circular convolutions. All models using finite linear convolutions perform significantly worse than the baseline and our NIFF. This observation is consistent with high-resolution data like ImageNet-100.

| Method | Top 1 Acc |
|---|---|
| ConvNeXt-tiny Liu et al. (2022b) | 90.37 |
| NIFF (ours) | **91.48** |
| NIFF linear | 84.30 |
| ResNet-18 He et al. (2016) | **92.74** |
| NIFF full (ours) | 92.66 |
| NIFF full linear | 85.10 |
| ResNet-18 separated conv | 90.18 |
| NIFF (ours) | 90.63 |
| NIFF linear | 83.11 |
| ResNet-50 He et al. (2016) | **93.75** |
| NIFF full (ours) | 93.39 |
| NIFF full linear | 88.22 |
| ResNet-50 separated conv | 92.13 |
| NIFF (ours) | 93.11 |
| NIFF linear | 87.54 |
| DenseNet-121 Huang et al. (2017) | **93.93** |
| NIFF full (ours) | 92.47 |
| NIFF full linear | 85.73 |
| DenseNet-121 separated conv | 92.00 |
| NIFF (ours) | 92.49 |
| NIFF linear | 79.95 |
| MobileNet-v2 Sandler et al. (2018) | **94.51** |
| NIFF (ours) | 94.03 |
| NIFF linear | 93.21 |

Table A5: Average training time per epoch in seconds and standard deviation on one NVIDIA Titan V of NIFF compared to standard spatial convolutions 3x3 or 7x7 and maximal larger spatial convolutions on CIFAR-10.

| Name | Baseline 3x3/7x7 | Spatial Conv featuremap sized | NIFF (ours) |
|---|---|---|---|
| ConvNeXt-tiny Liu et al. (2022b) | 68.31 ± 0.62 | 108.97 ± 0.25 | 96.48 ± 1.97 |
| ResNet-18 He et al. (2016) | 8.57 ± 0.20 | 22.75 ± 0.28 | 17.87 ± 0.54 |
| ResNet-50 He et al. (2016) | 7.98 ± 0.10 | 37.05 ± 0.48 | 27.36 ± 0.16 |
| DenseNet-121 Huang et al. (2017) | 26.36 ± 1.32 | 67.11 ± 0.25 | 84.51 ± 3.71 |
| MobileNet-v2 Sandler et al. (2018) | 22.50 ± 0.26 | 143.47 ± 0.20 | 83.41 ± 4.78 |

# F   Ablation on more modules

We show that our NIFF can be combined with other frequency modules to achieve networks that operate mostly in the frequency domain. Hence, the number of transformations can be reduced. Table A9 demon-

Table A6: Average training time per epoch in seconds and standard deviation on four NVIDIA A100 of NIFF compared to standard spatial convolutions (3x3 or 7x7) and maximal larger spatial convolutions on ImageNet-100.

| Name | Baseline 3x3/7x7 | Spatial Conv featuremap sized | NIFF (ours) |
|---|---|---|---|
| ConvNeXt-tiny Liu et al. (2022b) | $92.19 \pm 4.21$ | $487.89 \pm 3.54$ | $149.70 \pm 1.32$ |
| ResNet-18 He et al. (2016) | $31.99 \pm 0.71$ | $608.13 \pm 22.53$ | $89.00 \pm 0.60$ |
| ResNet-50 He et al. (2016) | $85.51 \pm 0.22$ | $951.43 \pm 6.09$ | $204.82 \pm 0.33$ |
| ResNet-101 He et al. (2016) | $152.39 \pm 5.07$ | $1392.21 \pm 47.91$ | $349.83 \pm 2.44$ |
| DenseNet-121 Huang et al. (2017) | $128.95 \pm 1.64$ | $10188.15 \pm 28.17$ | $408.08 \pm 2.20$ |
| MobileNet-v2 Sandler et al. (2018) | $32.64 \pm 0.25$ | $856.84 \pm 4.35$ | $100.75 \pm 0.15$ |

Table A7: Average inference time in seconds and standard deviation on one NVIDIA Titan V of NIFF compared to standard spatial convolutions (3x3 or 7x7) and maximal larger spatial convolutions on the full CIFAR-10 validation set.

| Name | Baseline 3x3/7x7 | Spatial Conv featuremap sized | NIFF (ours) |
|---|---|---|---|
| ConvNeXt-tiny Liu et al. (2022b) | $2.35 \pm 0.05$ | $4.06 \pm 0.02$ | $6.32 \pm 0.14$ |
| ResNet-18 He et al. (2016) | $2.69 \pm 0.10$ | $3.29 \pm 0.08$ | $3.14 \pm 0.54$ |
| ResNet-50 He et al. (2016) | $3.09 \pm 0.13$ | $5.05 \pm 0.45$ | $5.26 \pm 0.06$ |
| DenseNet-121 Huang et al. (2017) | $3.40 \pm 0.05$ | $5.27 \pm 0.02$ | $6.50 \pm 0.14$ |
| MobileNet-v2 Sandler et al. (2018) | $2.93 \pm 0.06$ | $8.68 \pm 0.43$ | $7.02 \pm 0.04$ |

strates that adding the downsampling layer Grabinski et al. (2022; 2023) or the last average pooling and the fully connected layer also yields good results. Also combining all of them, NIFF, FLC Pooling Grabinski et al. (2022) and the Average Pooling plus Fully connected layer performs quite well. Also, incorporating the ComplexBatchNorm Trabelsi et al. (2018) leads to a drop in accuracy by roughly 15%. We also tried to incorporate the non-linearity into the frequency domain, but we were not able to achieve much better results than by removing it fully.

## G  Ablation on Padding

We evaluate our NIFF when the featuremaps are padded with different kinds of padding methods. Due to the padding of the featuremaps and the cropping after the application of our NIFF, possible artifacts can be mitigated. The padding is applied around the featuremaps before transforming them into the frequency domain. The padding size is as large as the original featuremap. After the application of our NIFF, the featuremaps are transformed back into the spatial domain and cropped to their original size. The resulting networks experience similar performance as our baseline NIFF as shown in Table A10. Figure A18 and Figure A19 show the learned spatial kernels when only the featuremaps are padded. The learned spatial kernels are still relatively small and well-localized.

## H  NIFF's architecture

In the following, we describe the architecture used for our NIFFs for each backbone network architecture. Note that the size of the NIFF is adjusted to the size of the baseline network as well as the complexity of the classification task.

Table A8: Average inference time in seconds and standard deviation on four NVIDIA A100 of NIFF compared to standard 3x3 full or 7x7 depth-wise spatial convolutions and maximal larger spatial convolutions on the full ImageNet-100 validation set.

| Name | Baseline 3x3/7x7 | Spatial Conv featuremap sized | NIFF (ours) |
|---|---|---|---|
| ConvNeXt-tiny Liu et al. (2022b) | $4.86 \pm 0.13$ | $17.07 \pm 0.19$ | $6.06 \pm 0.08$ |
| ResNet-18 He et al. (2016) | $4.53 \pm 0.52$ | $24.57 \pm 0.21$ | $4.52 \pm 0.18$ |
| ResNet-50 He et al. (2016) | $5.43 \pm 0.30$ | $42.33 \pm 0.21$ | $10.35 \pm 0.14$ |
| ResNet-101 He et al. (2016) | $7.35 \pm 0.17$ | $54.85 \pm 0.25$ | $16.82 \pm 0.10$ |
| DenseNet-121 Huang et al. (2017) | $5.12 \pm 0.07$ | $104.36 \pm 0.40$ | $12.04 \pm 0.18$ |
| MobileNet-v2 Sandler et al. (2018) | $4.27 \pm 0.13$ | $28.97 \pm 0.04$ | $6.61\ 0.16$ |

Table A9: Comparison on CIFAR-10 of NIFF MobileNet-v2 Sandler et al. (2018) incorporating more modules besides our NIFF into the frequency domain.

| NIFF | FLC Pooling [2022] | Complex BatchNorm [2018] | AveragePooling + FC | Accuracy |
|---|---|---|---|---|
| ✓ | ✗ | ✗ | ✗ | **94.03** |
| ✓ | ✓ | ✗ | ✗ | 93.61 |
| ✓ | ✗ | ✓ | ✗ | 78.60 |
| ✓ | ✗ | ✗ | ✓ | 93.82 |
| ✓ | ✓ | ✓ | ✗ | 79.83 |
| ✓ | ✓ | ✗ | ✓ | 93.27 |
| ✓ | ✗ | ✓ | ✓ | 73.90 |
| ✓ | ✓ | ✓ | ✓ | 55.81 |

**Low-resolution task**  All networks trained on CIFAR-10 incorporate the same NIFF architecture. The NIFF consists of two stacked $1 \times 1$ convolutions with a ReLU activation function in between. The $1 \times 1$ convolution receives as input two channels, which encode the $x$ and $y$ coordinate as described in Figure 2. The $1 \times 1$ convolution expands these two channels to 32 channels. From these 32 channels, the next $1 \times 1$ convolution maps the 32 channels to the desired number of point-wise multiplication weights.

**High-resolution task**  For the networks trained on ImageNet-100 and ImageNet-1k the size of the neural implicit function to predict the NIFF is kept the same for each architecture respectively, while the size of the neural implicit function is adjusted to the network architecture to achieve approximately the same number of trainable parameters. Hence, the lightweight MobileNet-v2 model Sandler et al. (2018) and the small DensNet-121 Huang et al. (2017) incorporate a smaller light-weight neural implicit function to predict the NIFF, while larger models like ResNet He et al. (2016) or ConvNeXt-tiny Liu et al. (2022b) incorporate a larger neural implicit function. For simplicity, we define two NIFF architectures. One for the large models and one for the smaller, lightweight models.

For the smaller, lightweight models, the neural implicit function consists of three stacked $1 \times 1$ convolutions with one SiLU activation after the first one and one after the second one. The dimensions for the three $1 \times 1$ convolutions are as follows. We start with two channels and expand to eight channels. From these eight channels, the second $1 \times 1$ convolution suppresses the channels down to four. Afterwards, the last $1 \times 1$ convolution maps these four channels to the desired number of point-wise multiplication weights.

For the larger models, we used four layers within the neural implicit function for NIFF. The structure is similar to all NIFFs between each $1 \times 1$ convolution a SiLU activation function is applied. The dimensions for the four layers are as follows. First from two to 16 channels, secondly from 16 to 128 channels and

Table A10: Evaluation of top 1 and top 5 accuracies on ImageNet-100 for ResNet-18 with different kinds of padding.

| Name | Acc@1 | Acc@5 |
|------|-------|-------|
| ResNet-18 baseline | 87.52 | 97.50 |
| NIFF (ours) | 86.52 | 97.14 |
| NIFF zero padding | 87.00 | 97.54 |
| NIFF reflect padding | 86.64 | 97.24 |
| NIFF circular padding | 87.06 | 97.34 |

afterwards suppressed down from 128 to 32 channels. The last $1 \times 1$ convolution maps these 32 channels to the desired number of point-wise multiplication weights.

We show that the smaller NIFF size for the lightweight models does not influence the resulting performance. Thus, we train a lightweight MobileNet-v2 with larger NIFFs (similar size as the larger models). The results are presented in Table A11. We can see that the network does not benefit from the larger NIFF size. Hence, we assume that keeping the smaller NIFFs for the smaller, lightweight models can achieve a good trade-off between the number of learnable parameters and performance.

Table A11: Evaluation of top 1 and top 5 accuracies on ImageNet-100 for different NIFF sizes for the lightweight MobileNet-v2 Sandler et al. (2018).

| Name | Acc@1 | Acc@5 |
|------|-------|-------|
| MobileNet-v2 baseline | **84.94** | 96.28 |
| small NIFF | 83.72 | **96.40** |
| big NIFF | 83.82 | 96.32 |

**Low-resolution task** For all models trained on CIFAR-10 the NIFF architectures is kept the same. The neural implicit function consists of two stacked $1 \times 1$ convolutions with one ReLU activation in between. The dimensions for the two $1 \times 1$ convolutions are as follows. We start with two channels and expand to 32 channels. The second $1 \times 1$ convolution maps these 32 channels to the desired number of point-wise multiplication weights.

**Ablation on Separated Convolution** Further, we ablate our design choices to use separated depth-wise and $1 \times 1$ convolutions instead of full convolutions for efficiency. Hence, we train all ResNet and DenseNet networks with additionally separated convolutions (separated in depth-wise and $1 \times 1$ convolution) as well as our NIFF as full convolution. Tables 2, A1 and A2 show that using separated convolutions in the spatial domain performs slightly worse than the baseline but also reduces the amount of learnable parameters similarly to our NIFF. Using full convolutions in our NIFF leads to an increase in accuracy but also an increased amount of learnable parameters. Hence, we can see a clear trade-off between number of learnable parameters and accuracy.

# I Training Details

**ImageNet.** The training parameters and data preprocessing are kept the same for ImageNet-1k and ImageNet-100. For the training of each network architecture, we used the data preprocessing as well as the general training pipeline provided by Liu et al. (2022b). The training parameters for each individual

network are taken from the original papers provided by the authors ResNet He et al. (2016), DenseNet-121 Huang et al. (2017) ConvNeXt-tiny Liu et al. (2022b) and MobileNet-v2 Sandler et al. (2018).

**CIFAR-10.** For CIFAR-10 we used the same training parameter for all networks. We trained each network for 150 epochs with a batch size of 256 and a cosine learning rate schedule with a learning rate of 0.02. we set the momentum to 0.9 and weight decay to 0.002. The loss is calculated via LabelSmoothingLoss with label smoothing of 0.1 and as an optimizer, we use Stochastic Gradient Descent (SGD).

For data preprocessing, we used zero padding by four and cropping back to $32 \times 32$ and horizontal flip, as well as normalizing with mean and standard deviation.

**Computing Infrastructure** For training our models and the baseline we use NVIDIA Titan V and NVIDIA A100 GPUs. For the training on low-resolution data (CIFAR-10). We used one NVIDIA Titan V, depending on the model architecture and the convolution used (baseline, NIFF or large convolution) the training took between 15 minutes and 90 minutes. For the training on high-resolution data (ImageNet-100 and 1k) we used four NVIDIA A100 in parallel. The training time depends on the used model architecture and varies if we used the full ImageNet-1k dataset or only ImageNet-100. The training time for ImageNet-1k varies between one day and one hour and ten days and nine hours for ImageNet-100 between 93 minutes and one day eight hours dependent on the model architecture and the number of epochs for training.

## J   Convolution Theorem

Following, we demonstrate the proof of the convolution theorem. For more details, please refer to (for example) Bracewell & Kahn (1966); Forsyth & Ponce (2003).

As stated in Equation 1 in the main paper we make use of the convolution theorem Bracewell & Kahn (1966); Forsyth & Ponce (2003) which states that a circular convolution, denoted by $\circledast$, between a signal $g(x)$ and filter $k(x)$ in the spatial domain can be equivalently represented by a point-wise multiplication, denoted by $\odot$, of these two signals in the frequency domain, by computing their Fourier Transform, denoted by the function $\mathcal{F}(.)$:

$$\mathcal{F}(g \circledast k) = \mathcal{F}(g) \odot \mathcal{F}(k) \tag{7}$$

with

$$\mathcal{F}(g(x)) = G(u) = \int_{-\infty}^{\infty} g(x) e^{-j2\pi ux} dx \tag{8}$$

To show that this holds, we first show that the Fourier transformation **as a system** has specific properties when the signal is shifted. If we shift a signal/function $g(x)$ by $a$ in the spatial domain expressed by $g(x-a)$ this results in a linear phase shift in the Fourier domain:

$$\mathcal{F}(g(x-a)) = \mathcal{F}(g(x')) = \int_{-\infty}^{\infty} g(x') e^{-j2\pi u(x'+a)} dx' \tag{9}$$

where $e^{-j2\pi u(x'+a)} = e^{-j2\pi ua} e^{-j2\pi ux'}$ and $e^{-j2\pi ua}$ is a constant, such that

$$\mathcal{F}(g(x-a)) = e^{-j2\pi ua} G(u) \tag{10}$$

Using the shift property of the Fourier transform we can now prove the convolution theorem. The continuous convolution is defined as follows:

$$g(x) \circledast k(x) = \int_{-\infty}^{\infty} g(x) k(y-x) dx \tag{11}$$

The Fourier transformation of $g(x) \circledast k(x)$ is defined by:

$$\int_{-\infty}^{\infty} \left[ \int_{-\infty}^{\infty} g(x)k(y-x)dx \right] e^{-j2\pi uy} dy \tag{12}$$

By reversing the order of the integration we get

$$\int_{-\infty}^{\infty} g(x) \left[ \int_{-\infty}^{\infty} k(y-x)e^{-j2\pi uy} dy \right] dx \tag{13}$$

where we can pull out $g(x)$. Given the shift property, the inner integration can be defined by:

$$\int_{-\infty}^{\infty} k(y-x)e^{-j2\pi uy} dy = \mathcal{F}(k(y-x)) = e^{-j2\pi ux} K(u) \tag{14}$$

such that

$$
\begin{aligned}
&\int_{-\infty}^{\infty} g(x) \left[ \int_{-\infty}^{\infty} k(y-x)e^{-j2\pi uy} dy \right] dx \\
&= \int_{-\infty}^{\infty} g(x)e^{-j2\pi ux} K(u) dx \\
&= \left[ \int_{-\infty}^{\infty} g(x)e^{-j2\pi ux} dx \right] K(u) \\
&= G(u)K(u) = \mathcal{F}(g)(u)\mathcal{F}(k)(u),
\end{aligned}
\tag{15}
$$

so that for all spatial frequencies $u$, we have

$$\mathcal{F}(g \circledast k)(u) = \mathcal{F}(g)(u) \odot \mathcal{F}(k)(u). \tag{16}$$

## K    Fast Fourier Transform

The Discrete Fourier Transform (DFT) of an input signal $f(n)$ with $N$ samples is defined as

$$F(k) = \sum_{n=0}^{N-1} f(n)e^{-j2\pi kn/N} \tag{17}$$

Executing the DFT directly would take $O(N^2)$. Thus Cooley & Tukey (1965) developed the Fast Fourier Transform, short FFT. Which builds upon a divide and concur strategy and reduces the runtime down to $O(NlogN)$.

They used the inherent symmetry which results from the period nature of the transformed signal. To give an intuition for this inherent symmetry lets explore what happens if we shift by $N$:

$$
\begin{aligned}
F(k+N) &= \sum_{n=0}^{N-1} f(n)e^{-j2\pi(k+N)n/N}, \\
&= \sum_{n=0}^{N-1} f(n)e^{-j2\pi n}e^{-j2\pi kn/N}, \\
&= \sum_{n=0}^{N-1} f(n)e^{-j2\pi kn/N},
\end{aligned}
\tag{18}
$$

as $e^{j2\pi n} = 1$ for any integer $n$. Thus one can see that

$$F(k + N) = F(k) \tag{19}$$

and also

$$F(k + iN) = F(k) \tag{20}$$

for any integer $i$ holds.

Given this symmetry, Cooley & Tukey (1965) developed an algorithm which divides the DFT into smaller parts such that the DFT can be solved via divide and concur. Following we rearrange the DFT into two parts:

$$
\begin{aligned}
F(k) &= \sum_{n=0}^{N-1} f(n)e^{-j2\pi kn/N} \\
&= \sum_{m=0}^{N/2-1} f(2m)e^{-j2\pi k2m/N} \\
&+ \sum_{m=0}^{N/2-1} f(2m+1)e^{-j2\pi k(2m+1)/N} \\
&= \sum_{m=0}^{N/2-1} f(2m)e^{-j2\pi km/(N/2)} \\
&+ e^{-j2\pi k/N} \sum_{m=0}^{N/2-1} f(2m+1)e^{-j2\pi km/(N/2)}
\end{aligned}
\tag{21}
$$

Each part represents the even-numbered and odd-numbered values respectively. However, the runtime is still the same as each term consist of $O(N/2)N$ computations so in total still $O(N^2)$.

Luckily, this division into two parts can be continued in each part again. Hence, the range of $k$ is $0 \le k \le N$ while $m$ is now in the range of $0 \le m \le M$ where $M = N/2$. Thus, solving the problem only takes half of the computations as before, $O(N^2)$ becomes $O(M^2)$ where $M$ is half the size of $N$. As long as $M$ is even-valued, we can apply divide the problem in even smaller parts, applying the divide and concur strategy which in an recursive implementation takes only $O(NlogN)$.

## L   Code Base

Implementation code for our NIFF CNNs is provided at: `https://anonymous.4open.science/r/NIFF1528anonymous` and will be made publicly available upon acceptance.

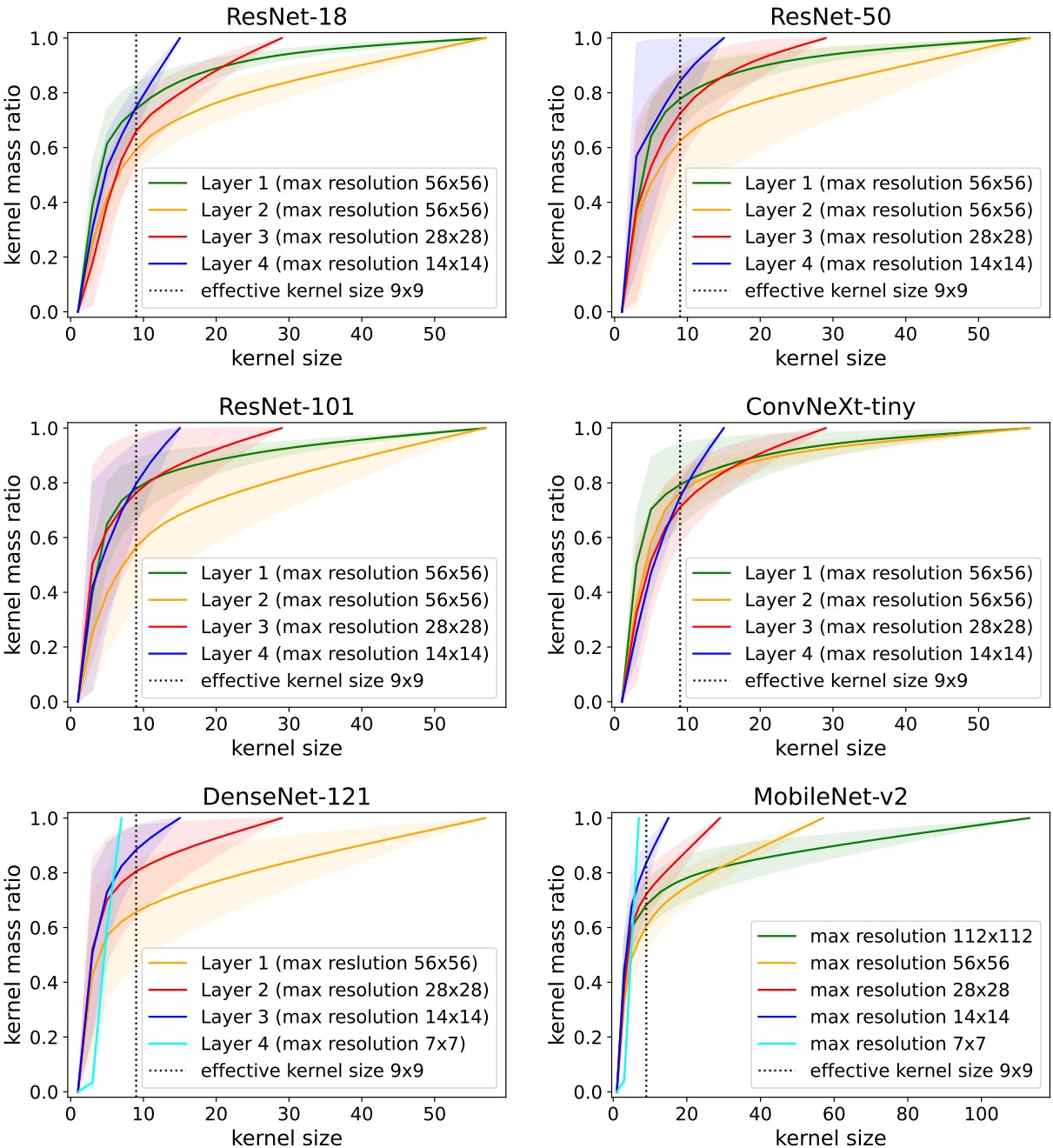

Figure A2: Effective kernel size evaluation on ImageNet-100. We plot the average ratio of the entire kernel mass contained within the limited spatial kernel size, where the x-axis denotes the width and height of the squared kernels. For ResNet and ConvNeXt-tiny each layer encodes one resolution. Thus, the layers could be summarised (Layer 1 encoding $56 \times 56$, Layer 2 $56 \times 56$, Layer 3 $28 \times 28$ and Layer 4 $14 \times 14$). For DenseNet-121 each layer can be summarised similarly, yet the after the first layer the feature maps are already downsampled resulting in the following: Layer 1 encoding $56 \times 56$, Layer 2 $28 \times 28$, Layer 3 $14 \times 14$ and Layer 4 $7 \times 7$. However, for MobileNet-v2 the resolution is downsampled within a layer.

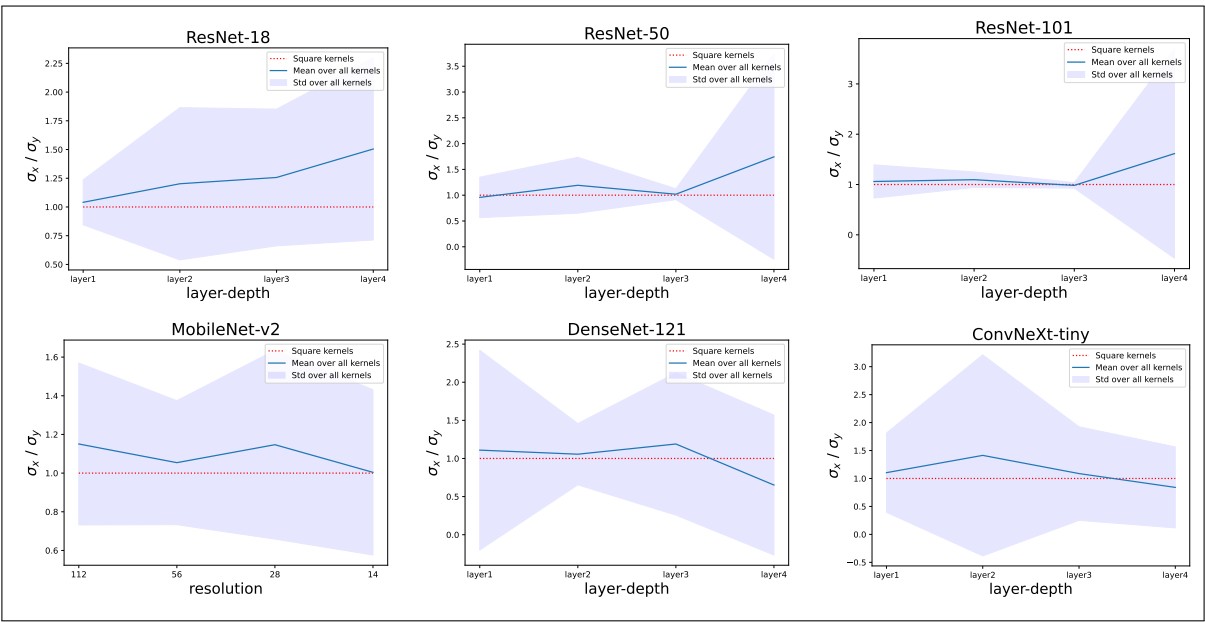

Figure A3: Analysis of non-square kernel shapes inspired by Romero et al. (2022a) on ImageNet-1k. We compare the variance $\sigma_x$ and $\sigma_y$ in x- and y-direction of a Gaussian fitted onto our learned spatial weights. The red dashed line indicates square-shaped kernels as the variance $\sigma_x$ and $\sigma_y$ are equal.

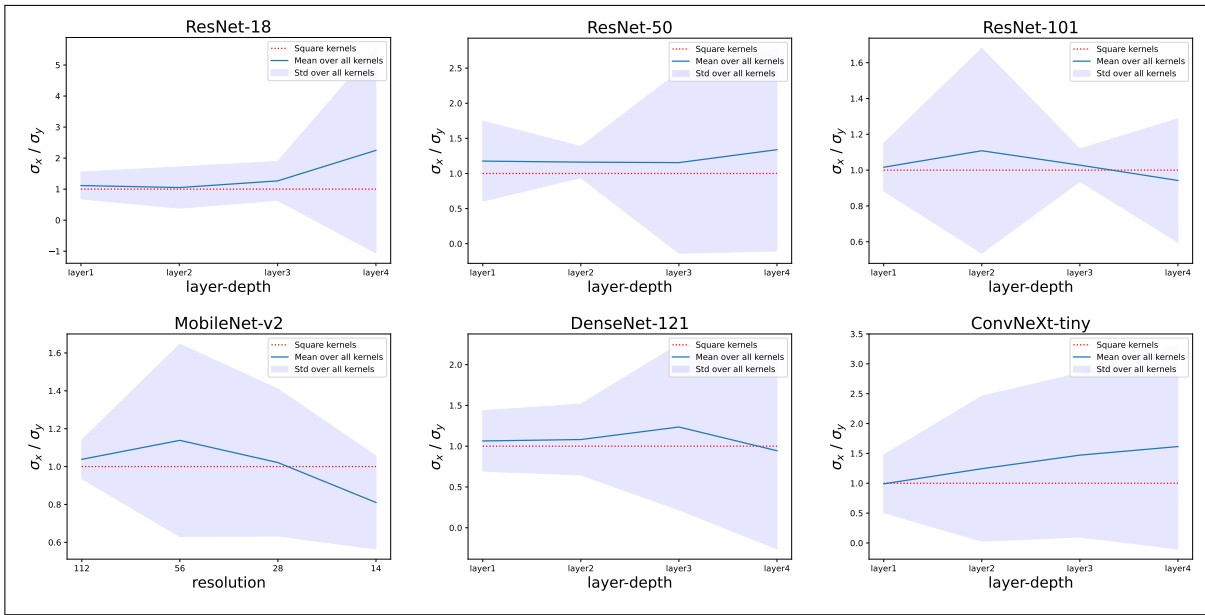

Figure A4: Analysis of non-square kernel shapes inspired by Romero et al. (2022a) on ImageNet-100. We compare the variance $\sigma_x$ and $\sigma_y$ in x- and y-direction of a Gaussian fitted onto our learned spatial weights. The red dashed line indicates square-shaped kernels as the variance $\sigma_x$ and $\sigma_y$ are equal.

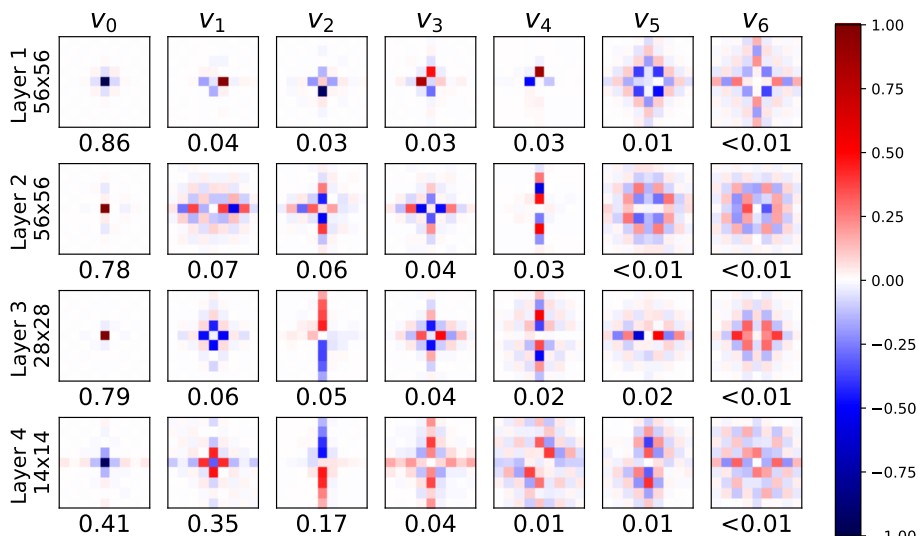

Figure A5: PCA basis and explained variance for each basis vector (below) of all spatial filters for each layer of a ConvNeXt-tiny trained on ImageNet-1k zoomed to $9 \times 9$. On the left, the maximal filter size for the corresponding layer is given. ConvNeXt convolutions are standardly equipped with larger kernel sizes than usual $(7 \times 7)$. However, our analysis reveals that the network barely uses large filters if it gets the opportunity to learn large filters. The learned filters in the first and third layer mostly use small $(3 \times 3)$, well-localized filters.

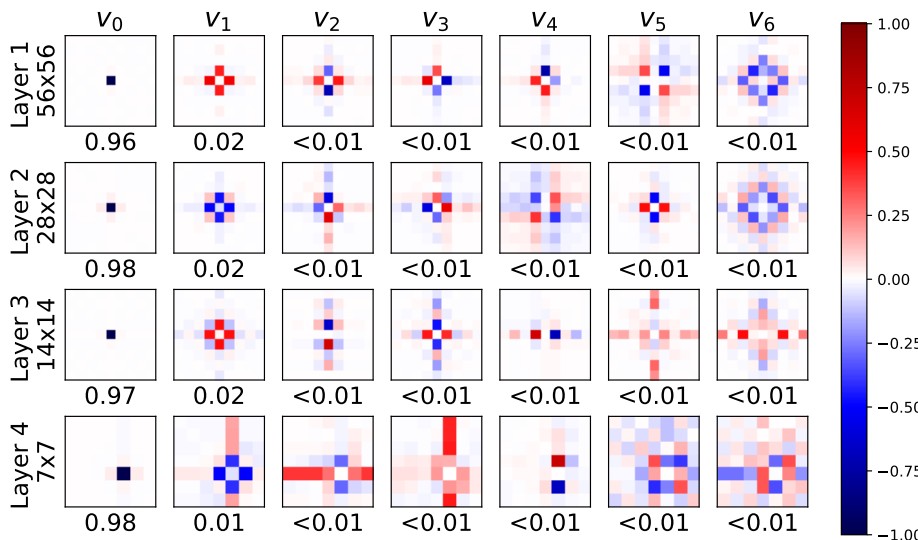

Figure A6: PCA basis and explained variance for each basis vector (below) of all spatial filters for each layer of a DenseNet-121 trained on ImageNet-1k zoomed to $9 \times 9$. On the left, the maximal filter size for the corresponding layer is given. We can see that most filters only use a well-localized, small kernel size although they could use a much bigger kernel.

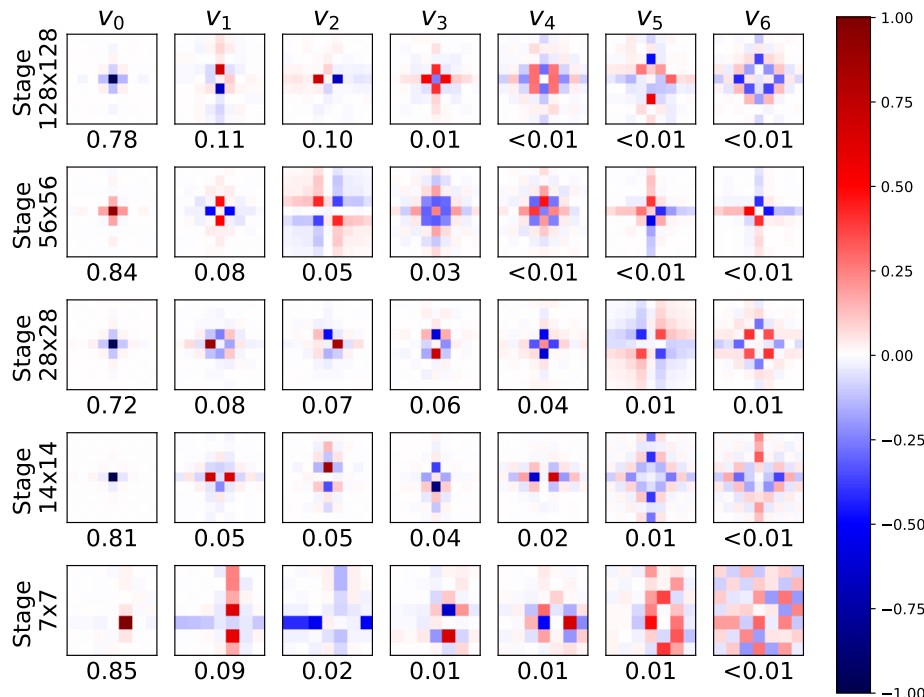

Figure A7: PCA basis and explained variance for each basis vector (below) of all spatial filters for each layer of a MobileNet-v2 trained on ImageNet-1k zoomed to $9 \times 9$. On the left, the maximal filter size for the corresponding stage is given. For MobileNet-v2 the feature maps are downsampled within a layer, thus the stages are combine by feature maps size rather than the layers. We can see that most filters only use a well-localized, small kernel size although they could use a much bigger kernel.

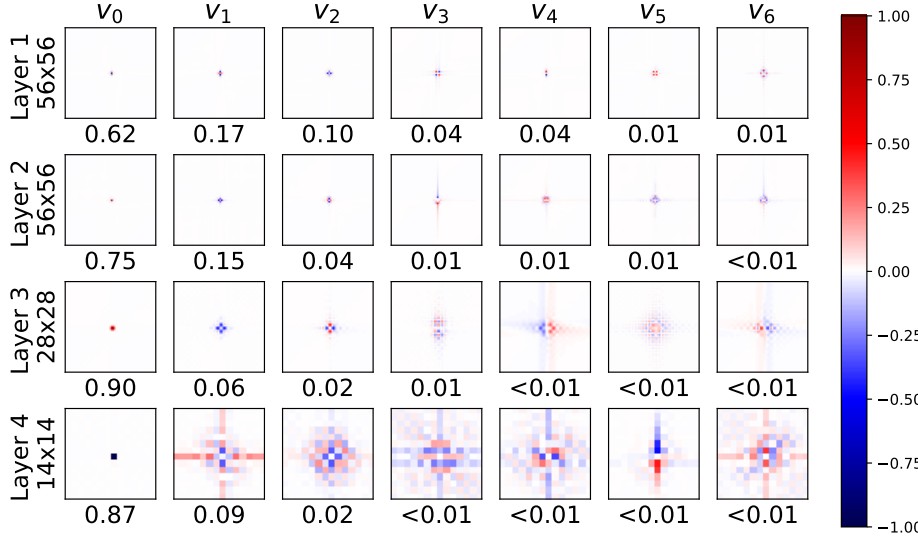

Figure A8: PCA basis and explained variance for each basis vector (below) of all spatial filters for each layer of a ResNet-50 trained on ImageNet-1k original size (not zoomed). On the left, the maximal filter size for the corresponding layer is given. We can see that most filters only use a well-localized, small kernel size although they could use a much bigger kernel.

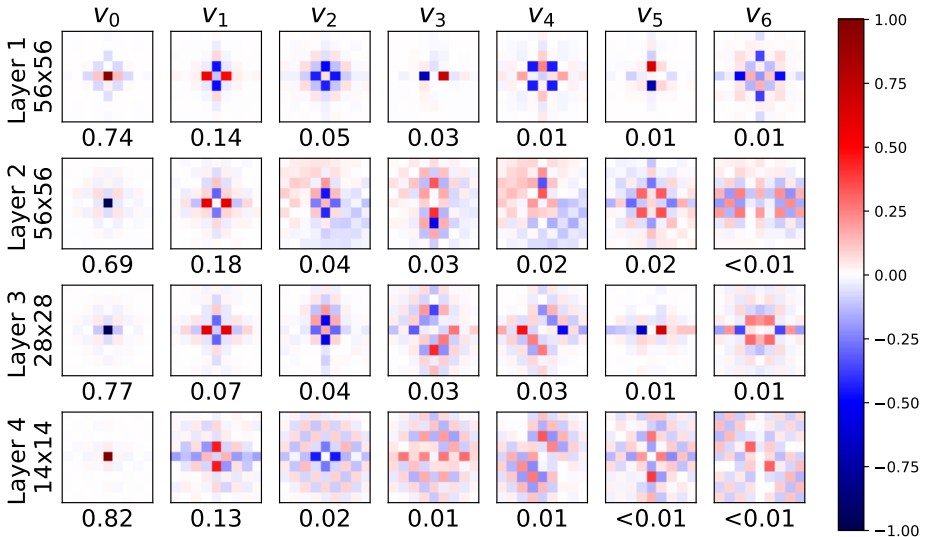

Figure A9: PCA basis and explained variance for each basis vector (below) of all spatial filters for each layer of a ResNet-50 trained on ImageNet-100 zoomed to $9 \times 9$. On the left, the maximal filter size for the corresponding layer is given. We can see that most filters only use a really small kernel size although they could use a much bigger kernel.

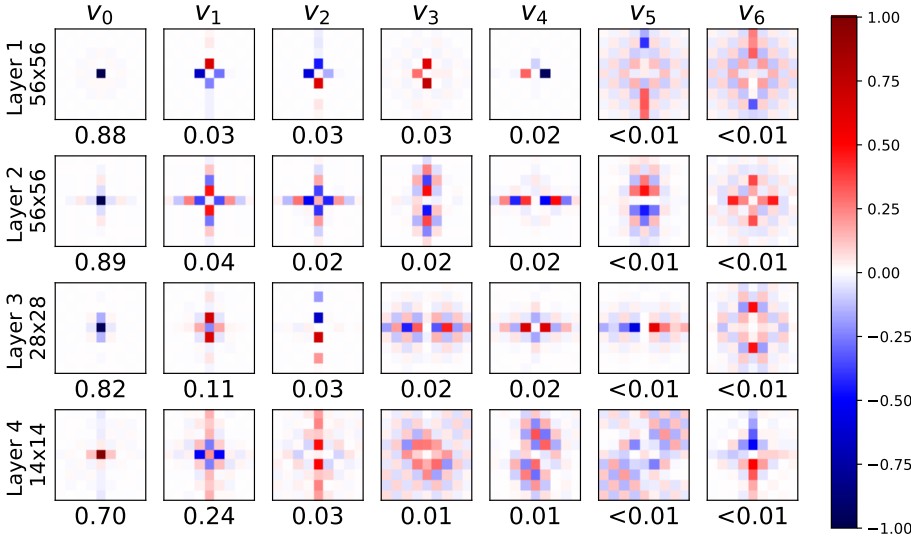

Figure A10: PCA basis and explained variance for each basis vector (below) of all spatial filters for each layer of a ConvNeXt-tiny trained on ImageNet-100 zoomed to $9 \times 9$. On the left, the maximal filter size for the corresponding layer is given. We can see that most filters only use a really small kernel size although they could use a much bigger kernel.

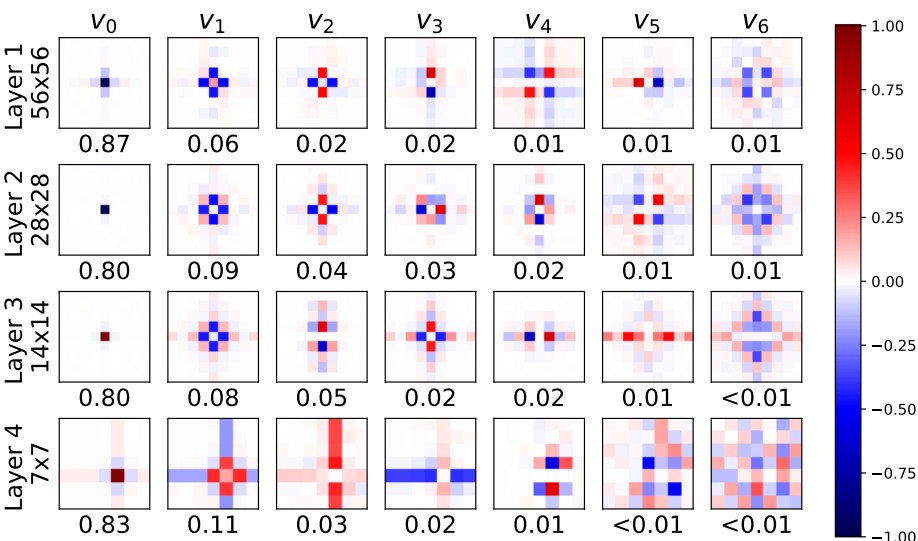

Figure A11: PCA basis and explained variance for each basis vector (below) of all spatial filters for each layer of a DenseNet-121 trained on ImageNet-100 zoomed to $9 \times 9$. On the left, the maximal filter size for the corresponding layer is given. We can see that most filters only use a well-localized, small kernel size although they could use a much bigger kernel.

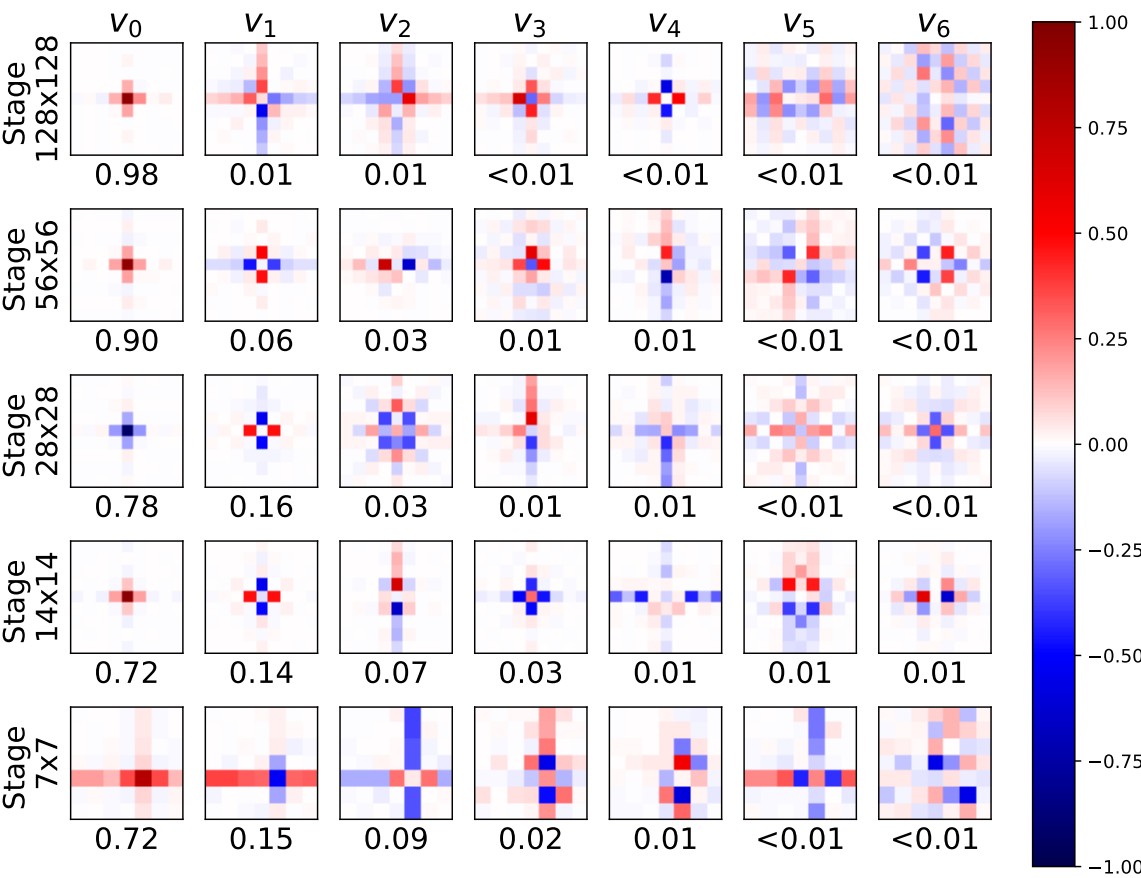

Figure A12: PCA basis and explained variance for each basis vector (below) of all spatial filters for each layer of a MobileNet-v2 trained on ImageNet-100 zoomed to $9 \times 9$. On the left, the maximal filter size for the corresponding stage is given. For MobileNet-v2 the feature maps are downsampled within a layer, thus the stages are combine by feature maps size rather than the layers. We can see that most filters only use a well-localized, small kernel size although they could use a much bigger kernel.

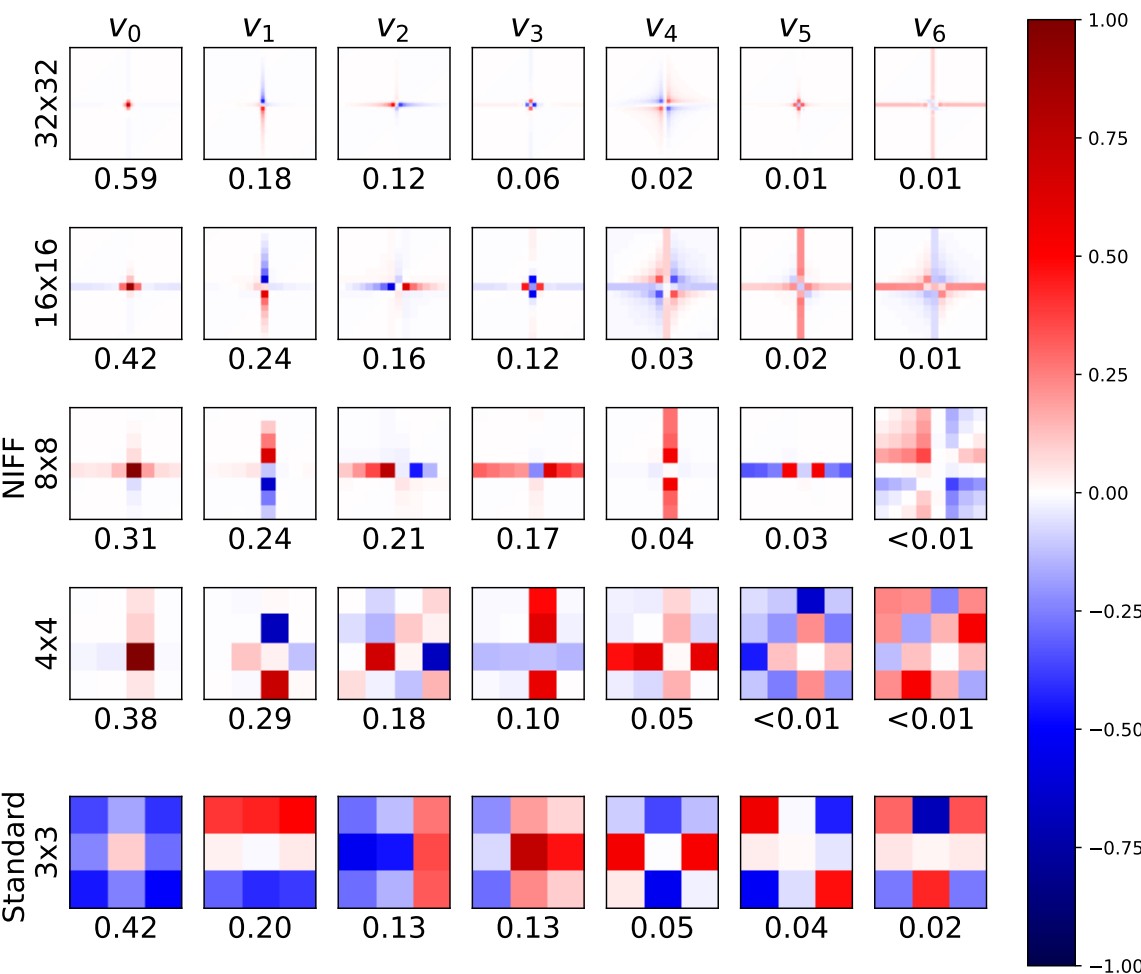

Figure A13: PCA basis and explained variance for each basis vector (below) of all spatial filters for each resolution for the NIFF convolutions of a MobileNet-V2 trained on CIFAR-10 as well as the learned filters for the third layer of a standard MobileNet-V2 trained on CIFAR-10 (bottom row). On the right, the maximal filter size for the corresponding layer is given. We can see that most filters only use a well-localized, small kernel size although they could use much bigger kernels.

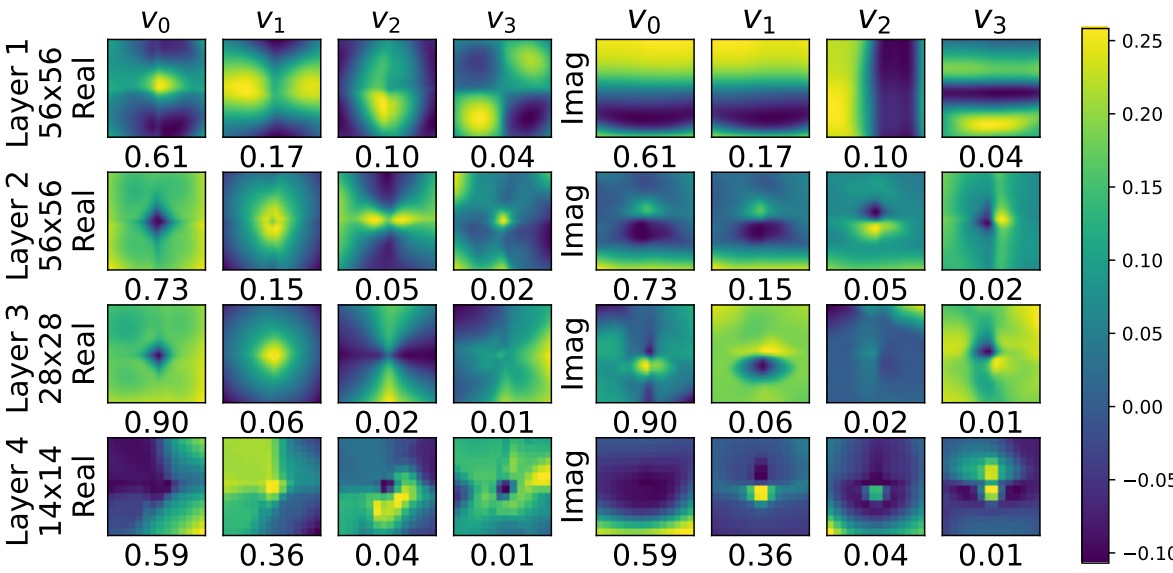

Figure A14: PCA basis and explained variance for each basis vector (below) of all element-wise multiplication weights for the real and imaginary part in the frequency domain for each layer of a ResNet-50 trained on ImageNet-1k. On the left, the maximal filter size for the corresponding layer is given. Right the weights for the real values are given, and on the left are the imaginary values.

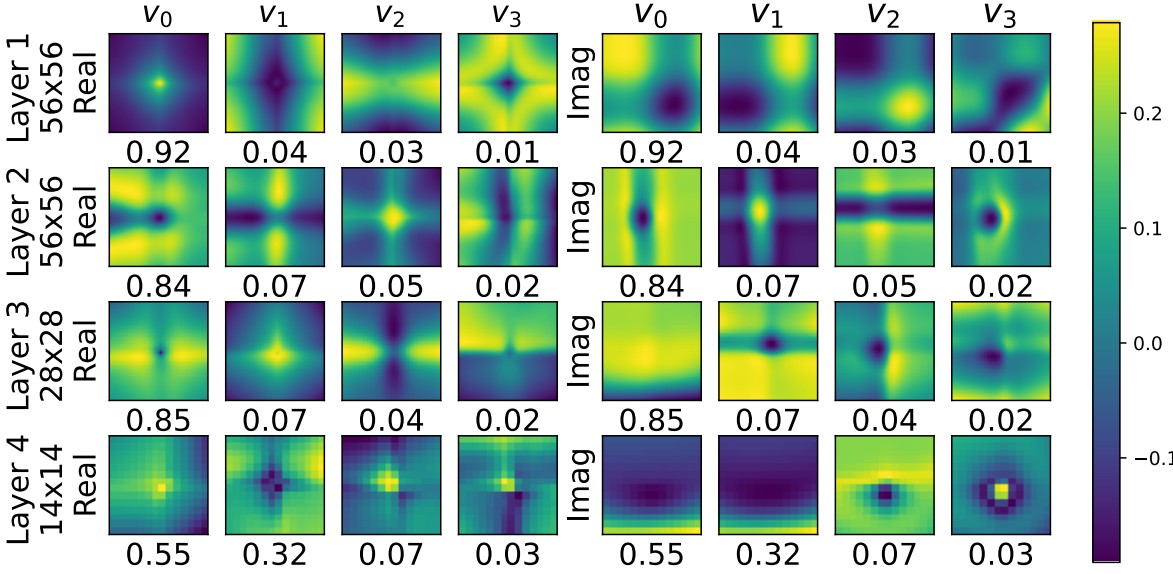

Figure A15: PCA basis and explained variance for each basis vector (below) of all element-wise multiplication weights for the real and imaginary part in the frequency domain for each layer of a ConvNeXt-tiny trained on ImageNet-1k. On the left, the maximal filter size for the corresponding layer is given. Right the weights for the real values are given, and on the left are the imaginary values.

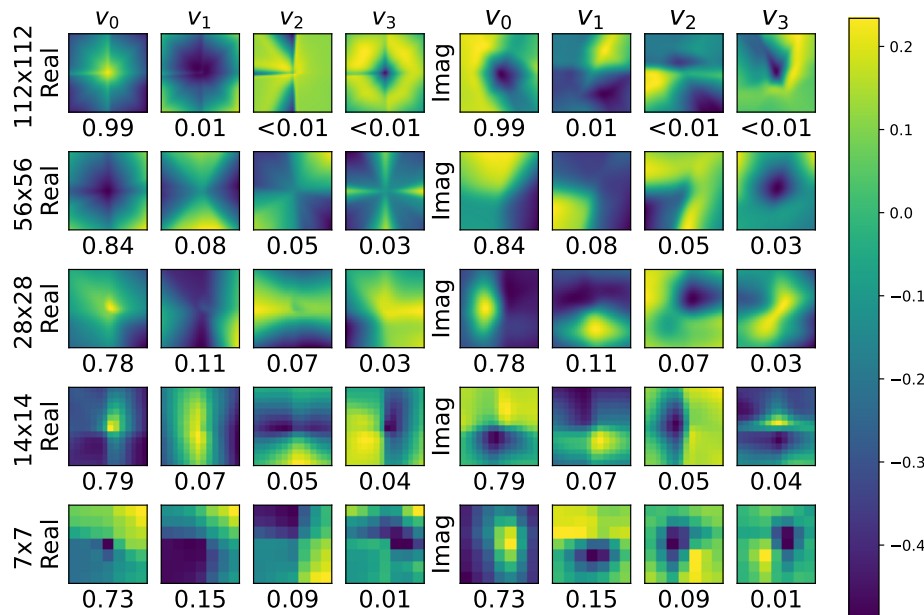

Figure A16: PCA basis and explained variance for each basis vector (below) of all element-wise multiplication weights for the real and imaginary part in the frequency domain for each layer of a MobileNet-v2 trained on ImageNet-1k. On the left, the maximal filter size for the corresponding layer is given. Right the weights for the real values are given, and on the left are the imaginary values.

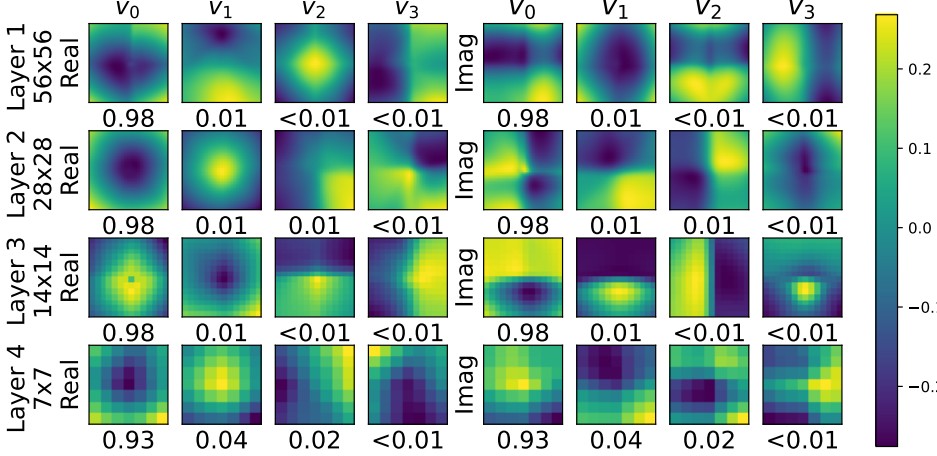

Figure A17: PCA basis and explained variance for each basis vector (below) of all element-wise multiplication weights for the real and imaginary part in the frequency domain for each layer of a DenseNet-121 trained on ImageNet-1k. On the left, the maximal filter size for the corresponding layer is given. Right the weights for the real values are given, and on the left are the imaginary values.

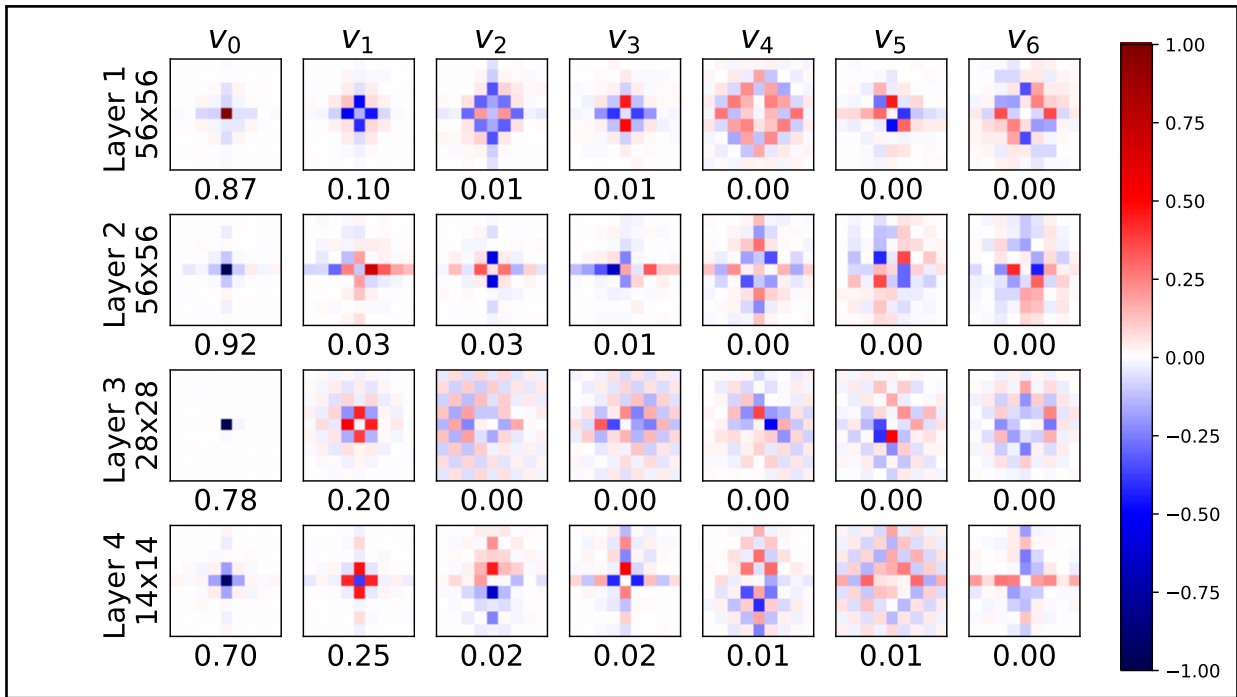

Figure A18: Actual kernels in the spatial domain of a ResNet-18 with additional zero padding before our NIFF trained on ImageNet-100. We plot for each kernel the zoomed-in (9 × 9) version below for better visibility. Still, most kernels exhibit well-localized, small spatial kernels.

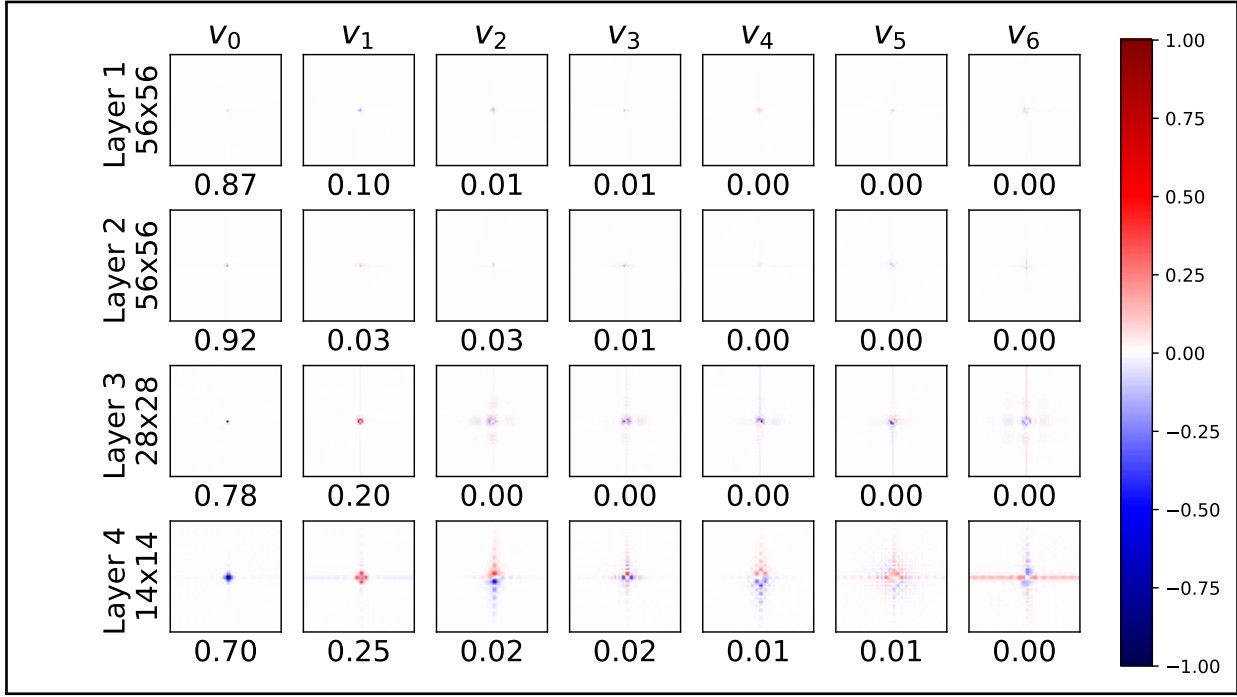

Figure A19: Actual kernels in the spatial domain of a ResNet-18 with additional zero padding before our NIFF trained on ImageNet-100. Still, most kernels exhibit well-localized, small spatial kernels.

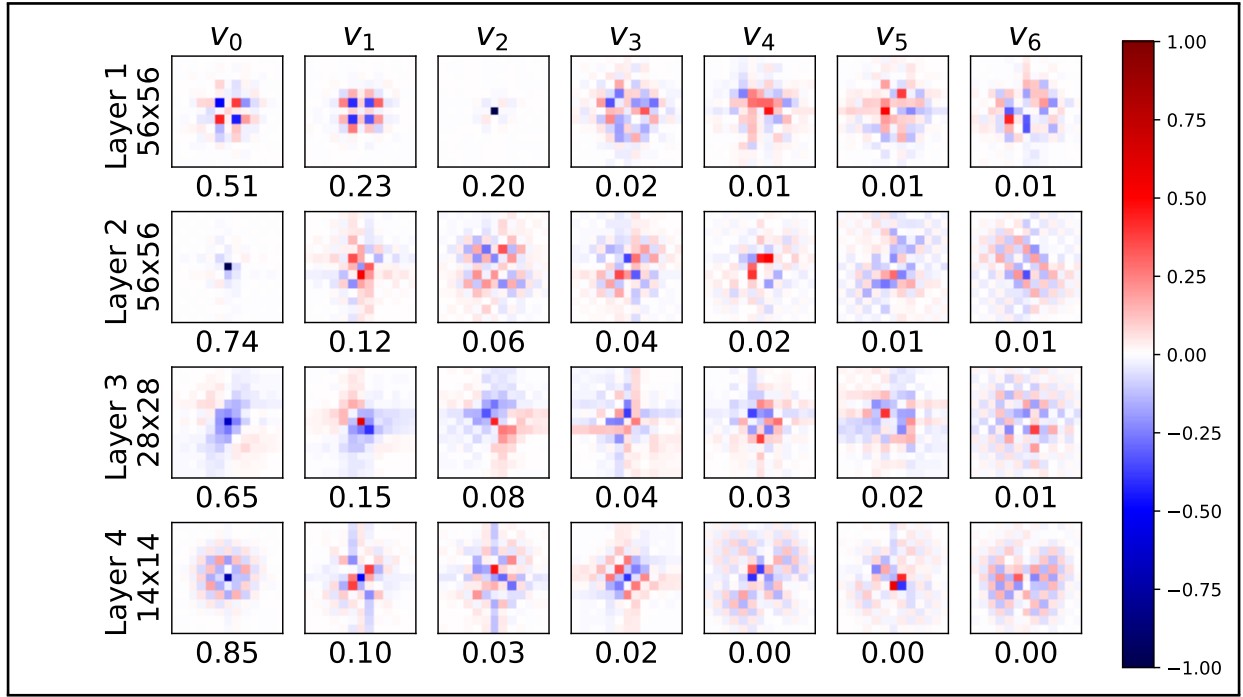

Figure A20: Actual kernels in the spatial domain of a ResNet-18 which mimics linear convolutions with our NIFF trained on ImageNet-100. We plot for each kernel the zoomed-in ($13 \times 13$) version below for better visibility. Still, most kernels exhibit well-localized, small spatial kernels. However, they are slightly larger than the kernels learned without padding and cropping.

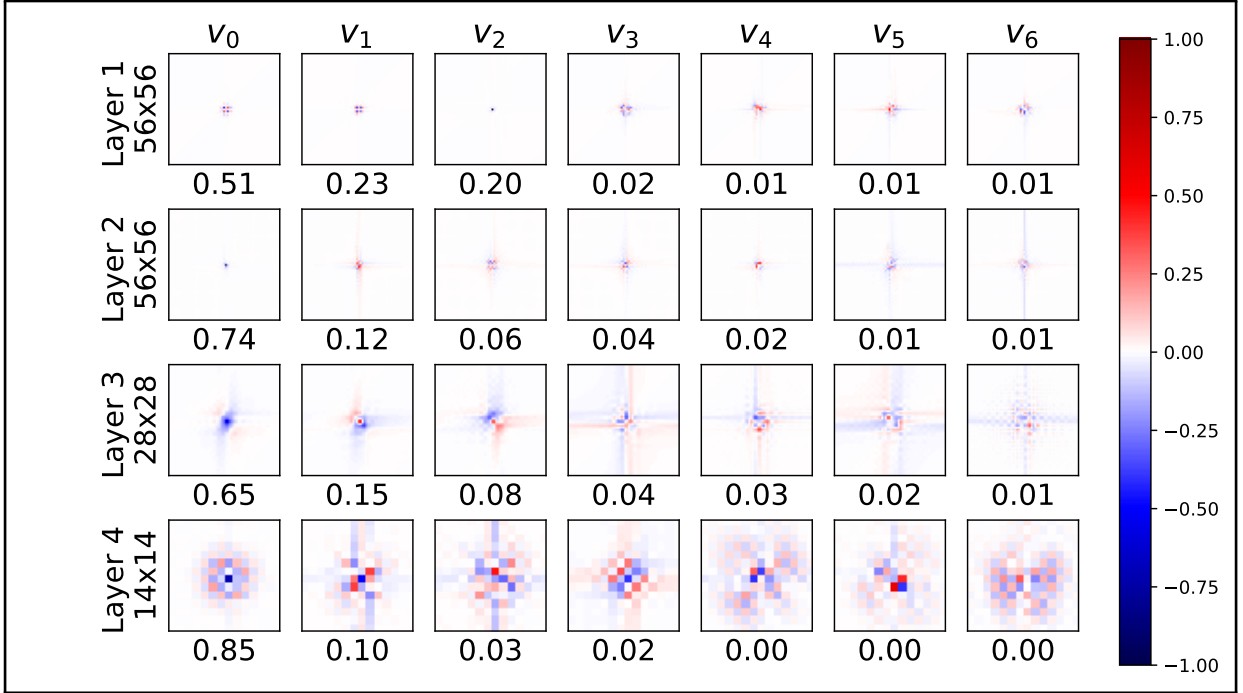

Figure A21: Actual kernels in the spatial domain of a ResNet-18 which mimics linear convolutions with our NIFF to mimic linear convolutions trained on ImageNet-100. Still, most kernels exhibit well-localized, small spatial kernels. However, they are slightly larger than the kernels learned without padding and cropping.

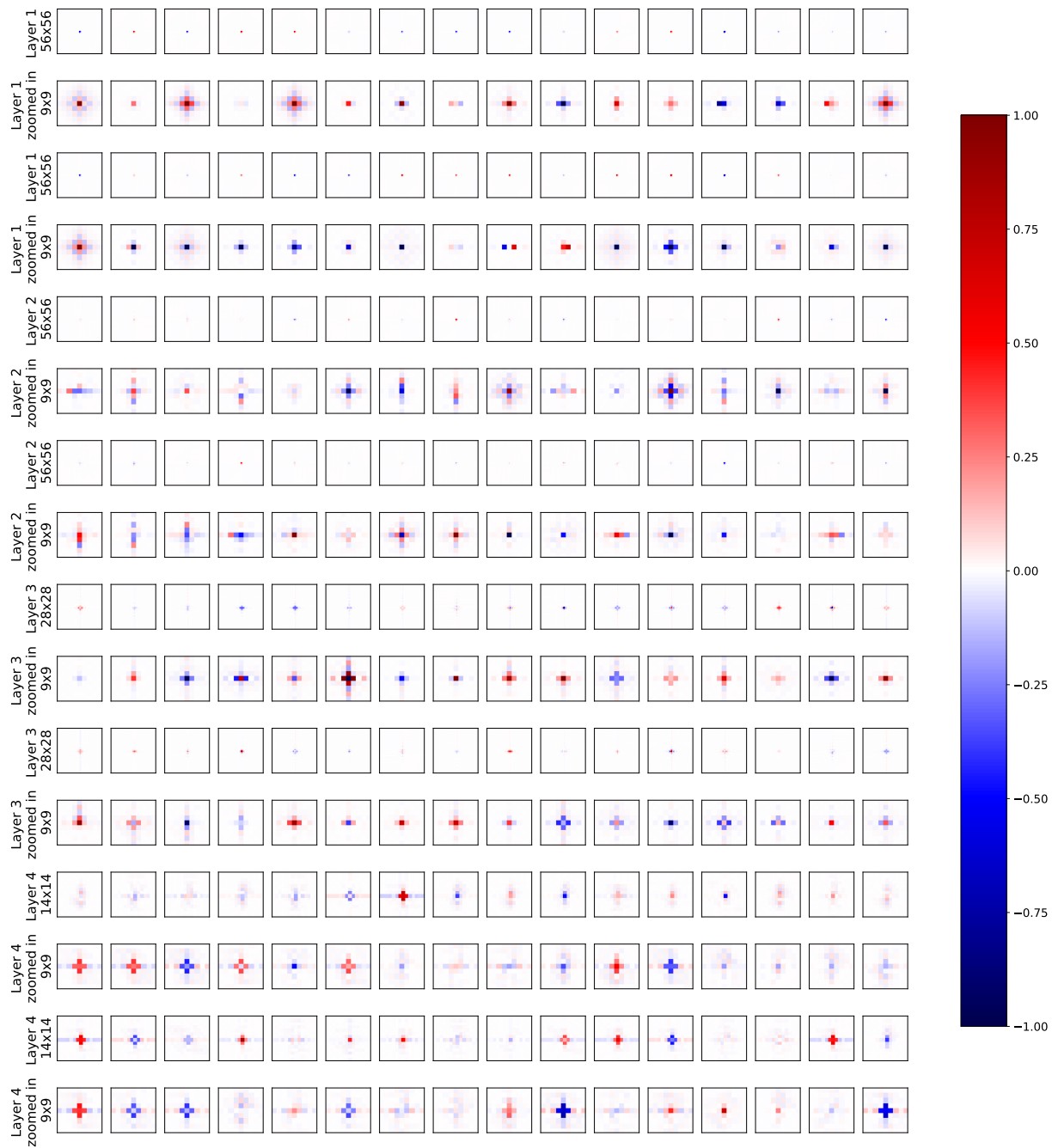

Figure A22: Actual kernels in the spatial domain of a ConvNeXt-tiny including our NIFF trained on ImageNet-1k. We plot for each kernel the zoomed-in $(9 \times 9)$ version below for better visibility. Overall, most kernels exhibit well-localized, small spatial kernels.

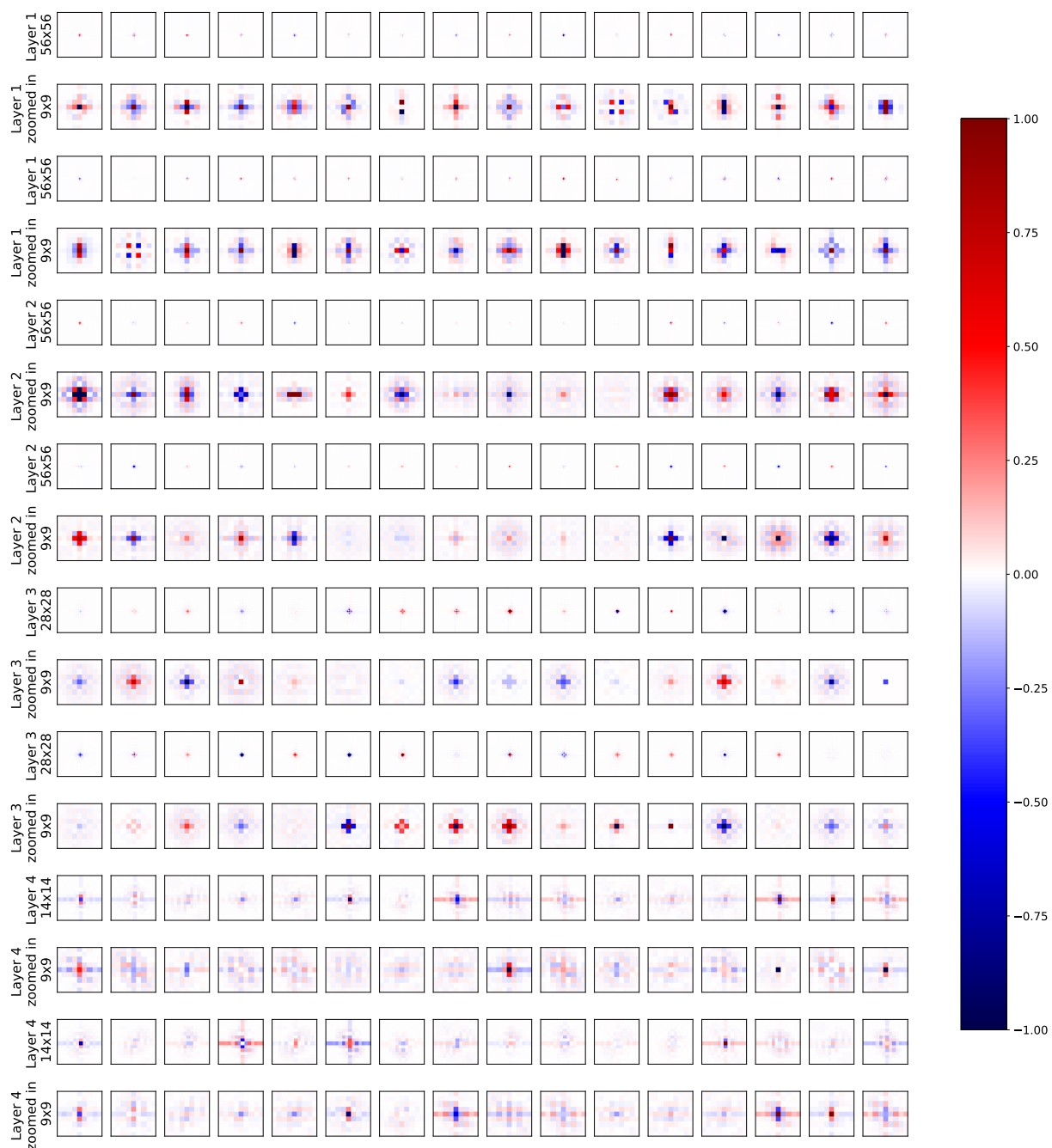

Figure A23: Actual kernels in the spatial domain of a ResNet-50 including our NIFF trained on ImageNet-1k. We plot for each kernel the zoomed-in $(9 \times 9)$ version below for better visibility. Overall, most kernels exhibit well-localized, small spatial kernels.

