# OpenReview forum: "As large as it gets – Studying Infinitely Large Convolutions via Neural Implicit Frequency Filters"
_TMLR — Accepted by TMLR_

### Review · Reviewer_4rXV · 2024-02-21

**Summary Of Contributions:**

This paper provides a study on the size of the optimal kernel size of different neural architectures. To this end, it proposes to parameterize conv kernels directly in the Fourier domain as an implicit neural representation, such that kernels of different sizes can be parameterized under the same number of parameters.

The study proposed in this paper is very interesting, as it concerns the family of long convolutional models, which have been becoming more popular in recent years.

**Audience:**

Yes

**Claims And Evidence:**

Yes

**Requested Changes:**

I believe that the study presented in this paper is important and relevant to the TMLR community. However, there are several points that convolute the message, the veracity and the conclusions of the paper. I believe that the paper requires a lot of rewriting and complementation before acceptance, which prevent me from directly recommending acceptance.

With that being said, I am open to support acceptance of the paper, should the authors address these concerns posed in the previous section during the discussion period.

**Strengths And Weaknesses:**

### Strengths

The paper outlines interesting findings, such as the fact that CNNs do not always seem to benefit from using global conv kernels even if this is possible. It also shows that basically all studied architectures can benefit from a more flexible parameterization than the usual 3x3, 7x7 considered in the baselines.

The experimental results are very appealing and encouraging for future research.

### Weaknesses

Unfortunately, I am afraid that there are several mixed messages in the paper, both in terms of findings and contributions, which weakens the novelty and validity of the approach as well as its findings.

* Papers that rely on long convolutions, e.g. CKConv, Hyena, S4, S5, just to name a few, all make use of the Fourier theorem to compute convolutions with very large kernels. The paper should make clear that this is part of the main pipeline considered in all existing long conv models. What is new is the fact that the kernels are directly parameterized in the Fourier domain.

* “At the same time, it remains unclear whether there is an optimal filter size and which size of filters would be learned could the model learn arbitrary sizes’
+-> As far as I understand, this is exactly the problem that FlexConv is trying to solve. In fact the findings found in that paper actually align very well with the findings obtained here. I know that there are different factors, but it should be clear that this has been done / tackled before.

* “First model to achieve results on par with state-of-the-art on standard high-resolution image classification benchmarks, while executing convs in the frequency domain.”
+->
On the related work section about Large kernel sizes, several important works are missing, e.g., CKConv, Hyena, S4, S5, etc. There’s an increasing body of literature on long convolutional models.
In Section 3, it is assumed that inputs are given in the Fourier domain. But this is clearly not the case, as only convolutions are performed in the Fourier domain. The rest of the operations are spatial. This should be very clear in the paper.
An important aspect that ties to the previous point is that the **CCNs trained in the frequency domain** considered here in comparisons try to solve a much more difficult problem. The main objective is to train and deploy CNNs **solely** on the Fourier domain, i.e., to construct networks on which only an FFT is used in the first layer, and an iFFT is used in the last layer of the network. Everything in between happens in the Fourier domain. This task is much more difficult than deploying convolutions on the Fourier domain, as this involves defining nonlinearities, normalization and other layers on the Fourier domain, which is very nontrivial. This difference should also be very clear in methodological and experimental comparisons.

* From what I understand, the only difference between the parameterization of CKConvs and NIFFs conv kernels is the fact that the latter is modeling the kernels directly on the frequency domain. I do not thing this is clear in Sec. 3.1. Also, in 3.1, it is not clear whether $\Phi$ is defined on complex numbers, or if this is real and only the output is made complex by the concatenation of two channels that correspond to real and imaginary parts. In which part does it become complex-valued? This is important, for instance, in terns of the nonlinearities used. In addition, it says here that the output is C-dimensional. But these values are complex. Depending on how this is parameterized, this could also be 2*C.

* In 3.2, there is a discussion on how Full-Convolutions are implemented. However, it is important to note that this is **not** a full convolution. In fact, what is explained here is the exact definition of depth-wise convolutions. Again, this should be clear and it should also be explained how this impacts the runtime, parameter count and accuracy.

* As explained previously the results in Tab 1 are misleading. AFAIK these architectures try to learn whole pipelines in the Fourier domain.
In 2D, long kernels could have rectangular forms, as shown in FlexConv. However, the metric used for the effective kernel evaluation only considers squared shapes. Have the authors considered this? I believe (based on the findings of FlexConv), that this could lead to misleading results. Also, it is clear that downsampling plays a major role on the effective kernel size of a layer. Has this been taken into account? For example, a kernel of size 14 in the 4th layer of ConvNext-tiny (Fig 4) is actually pretty much global in terms of the original resolution. This should be very clear in the paper to avoid misleading conclusions.

### Questions / Comments

* The paper makes use of circular convolutions on the Fourier domain, but it is also possible to define proper (non-circular) convolutions by zero padding the input. Is there a clear understanding of how this difference affects the results? For comparison, it could be possible to approximate this setting by training CNNs that use cyclic padding instead of zero padding. Or by padding feature maps pre FFT to avoid this.

* In FlexConv, the authors explicitly mention limitations related to the use of FFTConv. However, from what I understand, you did not experience these limitations. Could you comment on that?

---

> ### Author Response · Authors · 2024-03-23
> **Answer to your Review (Part1)**
>
> 1. Long Conv Related Work:
>
> Thank you for bringing up these works on sequence processing. We already distinguish between our work and CKConv [1] in our related work. Based on your suggestion, we added  Hyena [2] and S4 [3] to the related work, as they also make use of the convolution theorem for long convolutions. For S5 it appears that the authors do not make use of the convolution theorem. Rather they “replace[..] the frequency-domain approach used by S4 with a purely recurrent, time-domain approach leveraging parallel scans.” S5 [4]. We would appreciate it if the reviewer could specify in which context this work should be discussed - we will gladly add it.
>
> 2. Optimal filtersize and comparison to FlexConv [5]
>
> The goal of our work is to enable the individual analysis of the filter size for a given network architecture. In contrast, FlexConv [5] proposes to use a specific network architecture, FlexNets, to evaluate learned filter sizes. Furthermore, the parameterization in FlexConv, using Sirens, is significantly different from ours, leading to rather smooth regular filters in  FlexConv while NIFF filters learn smooth as well as non-smooth kernels (see for example in Figures A18 - A23).  We briefly discuss the relationship to FlexConv in Section 5  and have added a more extensive discussion in Appendix A. We also added Figures A3 and A4 to evaluate, similar to the evaluation in FlexConv, the aspect ratio of learned filters by fitting a Gaussian. However, since NIFFs do not necessarily yield spatially smooth kernels, the Gaussian fit should not be understood as a kernel mask for NIFFS as this is done in FlexConv.
>
> Our analysis of kernel sizes shows that there is no general, problem and architecture-independent, optimal filter size. In general, our findings are in general in line with the analysis in FlexConv - even though the filters look significantly different and we can easily consider a wider range of state-of-the-art architectures. We observe that higher resolution data (ImageNet) benefits from kernels larger than the typically used 3x3 kernels, while CIFAR-10 learns rather small filters. Similar to FlexConv, we also see that the filter sizes vary within one convolution. Hence, we might need to rethink the convolutional design overall. We added a more in-depth discussion on this aspect in Section 7.
>
> 3. CNN components in the frequency domain:
>
> We adapted Table 1 to be a purely related literature overview. We agree that a direct comparison to these previous approaches, which have been designed and optimized for various objectives, is not fair. None of the approaches (besides [1]) provide code to compare their approach. Hence, Table 1 is now part of the related work section and gives an overview of prior work that implements the convolution (or more network operations) in the frequency domain. Further, we list which architectures have been proposed and how well the networks perform on CIFAR-10 as most networks report this number. The figure caption makes clear that these numbers are not comparable - they just serve to provide a rough impression of the range of accuracies reached. Further, we added Table A9 to evaluate the combination of our approach (NIFF) with further operations in the frequency domain. Thus we show that we can include the downsampling via FLC Pooling [3], complex BatchNorm [4] and also global average pooling plus the fully connected layer with NIFF to train reasonably well-performing models. However, we find that, to keep up accuracy, we need to return to the spatial domain for the non-linearity, which is in line with the reviewers statement on the aim of previous papers: “ The main objective is to train and deploy CNNs solely on the Fourier domain [...] This task is much more difficult than deploying convolutions on the Fourier domain, as this involves defining nonlinearities, normalization and other layers on the Fourier domain, which is very nontrivial.”
>
> 4. Update of Section 3.1:
>
> We describe the general learning of kernels based on stacked 1x1 convolutions which receive two input featuremaps encoding the x and y directions in 3.1. In comparison, to CKConvs [1] we do not use a single coordinate, but we use the two coordinate-featuremaps as input. Further, CKConvs are executed in the spatial domain as long as the sequence is smaller than 50 samples. For our NIFF we learn the 2D filters directly in the frequency domain.
>
> 5. NIFF parameterization:
>
> We learn two independent networks for the real and imaginary parts and combine both to one complex representation. Thus, each network has C output channels. In Section 3.1, we provide the mathematical definition of the function mapping onto C complex numbers. To improve clarity, we have now specified in this section that our practical implementation parameterizes the prediction of real and imaginary parts of the filters separately.

---

> > ### Author Response · Authors · 2024-03-23
> > **Answer to your Review (Part2)**
> >
> > 6. Full Convolution :
> >
> > ConvNeXt and MobileNet do not incorporate full convolutions. ResNet und DenseNet do. Thus, we included an evaluation of the influence of splitting the full convolution into depthwise and 1x1 convolution in Table 2, A1 and A2. The resulting network shows very similar performance while having roughly as many parameters as NIFF.
> >
> > 7. Non-square-shaped convolutions and downsampling:
> >
> > a) Square shape of our evaluation: For the evaluation of the effective filter size in the spatial domain we used a square due to comparability to the standard convolutions applied in CNNs. Inspired by FlexConv [5], we now added an analysis of the variance in x and y direction for fitting a Gaussian onto our kernels in the spatial domain. Figures A3 and A4 demonstrate that most kernels are squared-shaped as the ratio between the variances in x and y is mostly one or near one. Yet, some kernels seem to also be rectangular and vary strongly as demonstrated in the standard deviation. We discuss these findings in section A in the appendix.
> >
> > b). Downsampling: The learned filter size in each layer is comparably small given the potential size. Yet, indeed, the receptive field is much larger when we use a kernelsize of 14x14 in the 4th layer. We added more details and intuition for this topic in Section 6.
> >
> > 8. Non-circular, linear convolutions with our NIFF:
> >
> > We ablate on padding the learned filters in the spatial domain to mimic linear, non-circular convolutions. We learn the filters in the frequency domain. However, for linear convolutions, they are transformed into the spatial domain and add zero padding.  This happens for featuremaps accordingly. Afterwards, filters and featuremaps are both transformed into the frequency domain to apply the point-wise multiplication. After back-transforming the resulting featuremaps into the spatial domain, they are cropped to the original size. This is obviously not particularly efficient. It merely serves the purpose of ablating the effect of linear versus circular kernels. The resulting models drop in accuracy consistently on ImageNet-100 and CIFAR-10 (Table A3 and Table A4). Hence comparing the learned filters learned with this method is questionable as their performance is not on par with the baseline.
> >
> > Additionally, we added experiments padding only the input prior to our NIFF. The resulting networks perform similarly to the networks without padding (Table A10) and the learned filters by our NIFF also exhibit a similar structure as the baseline without padding shown in Figures A18 and A19. Yet, padding the input for our NIFF would still result in the use of circular convolutions. We tried to mimic non-circular convolutions by padding the learned filters by our NIFF in the spatial domain. Yet, as mentioned above the resulting networks experience a consistent drop in accuracy and are thus not comparable to the baseline anymore.
> >
> > 9. Comparision to limitations in FlexConv [5]:
> >
> > There are two limitations mentioned in FlexConv, yet we believe your question is regarding the the following one:
> > Dynamic kernel sizes: computation and memory cost of convolutions with large kernels: We perform the convolutions directly in the frequency domain as pointwise multiplications. In Figure 8, we show the number of FLOPs for large convolutions (as large as the featuremaps) already exceeds the amount needed for our NIFF after a size of 32x32. Thus, our computational costs are reasonably small due to the use of our hypernetwork. Further, we only need to store the fixed amount of parameters for our hypernetworks as well as the input featuremaps encoding the x and y coordinates. Hence, we also have no issues with memory costs.
> >
> > [1] David W. Romero, et al. “CKConv: Continuous kernel convolution for sequential data.” In International Conference on Learning Representations, 2022
> >
> > [2] Poli, Michael, et al. "Hyena hierarchy: Towards larger convolutional language models." International Conference on Machine Learning. PMLR, 2023.
> >
> > [3] Albert Gu, et al.  “Efficiently modeling long sequences with structured state spaces.” In International Conference on Learning Representations, 2022
> >
> > [4] Jimmy T.H. Smith, et al. “Simplified State Space Layers for Sequence Modeling” The Eleventh International Conference on Learning Representations, 2023
> >
> > [5] Grabinski, Julia, et al. "Frequencylowcut pooling-plug and play against catastrophic overfitting." European Conference on Computer Vision. Cham: Springer Nature Switzerland, 2022.
> >
> > [6] Chiheb Trabelsi, et al. “Deep complex networks.” In International Conference on Learning Representations, 2018

---

> > > ### Comment · Reviewer_4rXV · 2024-03-28
> > >
> > > Dear authors,
> > >
> > > Thank you very much for your response. All of my concerns have been addressed in a high level of detail. I appreciate that very much.
> > >
> > > - Wrt 1. I apologize for my oversight. The current version is more than sufficient.
> > >
> > > I have a few questions left regarding the parameterization of the NIFF (there's no need to ablate more or anything. I am just curious about your thoughts and decision process for the current parameterization):
> > > - Is there any motivation behind using two independent networks for the real and imaginary part of the kernels? Wouldn't it be better to have a shared representation, as all channels could reuse the basis functions that the main network learns?
> > > - Also, it is well known that the real and imaginary part of complex representations are highly correlated (they parameterize a vector on the unitary circle). Therefore, I wonder if it wouldn't better to parameterize the complex components based on their magnitude and phase --either with two independent networks, or a shared one. For the magnitude and phase, simple abs() and mod -pi, pi operations would be sufficient to make sure that these values are always in the right range.
> > > - The NIFF uses normal nonlinearities, which suffer from an spectral bias. I wonder if using a "proper" neural field on the Fourier domain would have any added value.

---

> > > > ### Author Response · Authors · 2024-04-03
> > > >
> > > > We thank the reviewer for the positive feedback on our revision as well as for the interesting points of discussion! In the following, we will discuss these remaining questions.
> > > >
> > > > 1. In initial empirical experiments, we found that separating the networks leads to slightly better performance. Intuitively, we think that this should at least not be a disadvantage as the prediction of both, real and imaginary parts, depend on the same input values (the frequency coordinates) and can therefore learn proper mutual dependencies even when trained without shared parameters.
> > > >
> > > > 2. We mainly followed our intuition along three points that speak slightly against the parameterization in terms of magnitude and phase: (1) Parametrizing on magnitude and phase might require two very different parameterizations as the phase lies in a tangent space while the magnitude does not. (2) Further, while the magnitude is often quite intuitive and even somewhat correlated in the spatial frequencies, the phase is usually not, which is why we would expect it to be more difficult to learn. (3) A parameterization in magnitude and phase would require an additional pointwise transformation before applying the filter, which slightly adds to the overall cost.
> > > > Yet, we did not run any experiments with this setting, so it might be that it actually performs very well in practice and might be considered for future work.
> > > >
> > > >
> > > > 3. We assume you are referring to Siren networks? In empirical experiments, we found that using a sine activation function leads to a significant decrease in performance.
> > > >
> > > >
> > > > Please let us know if there are any further points you would like us to elaborate on!

---

> > > > > ### Comment · Reviewer_4rXV · 2024-04-03
> > > > >
> > > > > Dear authors,
> > > > >
> > > > > Thank you for the additional information.
> > > > >
> > > > > Wrt the last point, yes, I mean something like a SIREN or an MFN. The reason I am curious about this, is that such parameterizations allow for "non-smooth" representations. But this is perhaps not important in the Fourier domain, as frequencies are already packed together via the FFT. Do you agree with this? Do you have any idea as of why SIRENs worked worse in practice? I'd love to hear your thoughts about that.
> > > > >
> > > > > Best,
> > > > >
> > > > > Reviewer 4rXV

---

> > > > > > ### Author Response · Authors · 2024-04-04
> > > > > >
> > > > > > Again, thank you for your feedback - this discussion definitely inspires us to think about follow-up works. Regarding your last question, we conclude from our own work and current literature that sinusoidal nonlinearities or Fourier features in positional encodings allow us to efficiently approximate highly non-smooth functions in the spatial domain while this is hard with a relu activation, which only allows piecewise linear approximations. However, since we are considering relu-like activations in the Fourier domain, each of the resulting values at spatial frequency “positions” already represents a scaled sine or cosine wave. Therefore, when considering our learned kernels transformed to the spatial domain (please refer to Figures A22 and A23)  they turn out to be highly non-smooth in practice. We guess this is in line with your expectation “ But this is perhaps not important in the Fourier domain, as frequencies are already packed together via the FFT”?
> > > > > > As to why SIRENs worked worse in practice, our answer would be highly speculative. We have been considering our full training pipeline without explicitly initializing the NIFF to any specific values. One guess would be that SIRENs and comparable modules benefit from an informed initialization [7].
> > > > > > One last point is that by using sinusoidal activations, the filters - transformed to the spatial domain - are a combination of sine and cosine waves applied on a linear combination of sine and cosine waves. In the best case, this would allow for a quick change between the scale of neighboring frequencies. Yet, it might also require more training signal (higher number of iterations, more training data). Again, this is highly speculative and would obviously be an interesting case study for future work.
> > > > > >
> > > > > > Best,
> > > > > >
> > > > > > the authors
> > > > > >
> > > > > > [7] Sitzmann, V., Martel, J., Bergman, A., Lindell, D., & Wetzstein, G. (2020). Implicit neural representations with periodic activation functions. Advances in neural information processing systems, 33, 7462-7473.

---

> > > > > > > ### Comment · Reviewer_4rXV · 2024-04-04
> > > > > > >
> > > > > > > Dear Authors,
> > > > > > >
> > > > > > > Thank you for sharing your thoughts. Very interesting!
> > > > > > >
> > > > > > > Best,
> > > > > > >
> > > > > > > Reviewer 4rXV

---

### Review · Reviewer_Mxbe · 2024-03-01

**Summary Of Contributions:**

This paper proposes a way to build infinitely large convolutions via Neural Implicit Frequency filters. By transferring the learning from the spatial domain to the frequency domain, the proposed approach can reduce the learning complexity from O(N^4) to O(N^2log(N)). Using this technique, they can inpressively scale infinite convolutions to more complex dataset such as ImageNet-1K and ImageNet-100 with very good performance.

**Audience:**

Yes

**Broader Impact Concerns:**

No broader impact statement is provided.

**Claims And Evidence:**

Yes

**Requested Changes:**

Overall, I like the paper a lot. If the major weaknesses can be addressed properly, I will recommend accepting the paper.

**Strengths And Weaknesses:**

**Strengths:**

1. This paper proposes a principles way to train CNNs with infinite kernel sizes from the perspective of the frequency domain, which is another success of the math-inspired approach in deep learning.

2. The proposed approach has great potential to reduce the complexity of infinite kernels. The performance demonstrated in Table 2 is quite strong.

3. The analysis of effective kernel size is interesting.

**Weaknesses**

Major:

1. My biggest concern is the comparisons made in Table 1 and Table 2 are not very fair. In Table 1, the authors compare different frequency-domain approaches using different architectures, which is not very informative to see if the performance gains are obtained by NIFF or the advanced architectures. In Table 2, while using the same architectures, it is not very clear if models are trained exactly using the same configurations or not. After checking the appendix, I realized the models of NIFF are trained with more advanced recipes used by ConvNeXt, which is much stronger than the original one used by ResNet (He et al. 2016). I believe the performance of ResNet in Table 2 is not trained with the same recipes as NIFF, given its relatively low accuracy of 76.1% for ResNet50. Such an unfair comparison again is insufficient to justify the benefits of NIFF. What makes more sense to me is comparing different frequency-domain approaches with exactly the same training recipes and architectures.

2. Besides accuracy, it would be better to report training and inference costs to obtain the final model, as well as the model size including the hypernetworks.

3. Figure 8 only reports the theoretical FLOPs. While I understand it might be out of the scope of this paper for efficient implementation of NIFF, reporting the real training time will give a full picture to the readers. Although in this case, we might not obtain any speedups, I will not see this as a drawback.

Minor:

a. figure 2 and figure 3 are not very straightforward to understand the algorithm. I suggest modifying them to improve the presentation of the paper. b. can authors explain why DenseNet-121 NIFF and ConvNeXt-tiny NIFF fall short of the original models significantly? c. some papers appear repeatedly in the reference such as Imagenet: A large-scale hierarchical image database.

---

> ### Author Response · Authors · 2024-03-23
> **Answer to your Review**
>
> Updated Tables 1 and 2:
>
> We adapted Table 2 and added results for all baseline networks trained with the advanced data preprocessing provided by [2] and reported these numbers additionally to the ones reported by Pytorch for the timm model zoo. Our models are on par with the baseline. Interestingly, some models benefit from the improved augmentation pipeline while others (in particular MobileNet) rather show decreased accuracies.
>
> We adapted Table 1 to be a purely related literature overview. We agree that a direct comparison to these previous approaches, which have been designed and optimized for various objectives, is not fair. None of the approaches (besides [1]) provide code to compare their approach. Hence, Table 1 is now part of the related work section and gives an overview of prior work that implements the convolution (or more network operations) in the frequency domain. Further, we list which architectures have been proposed and how well the networks perform on CIFAR-10 as most networks report this number. The figure caption makes clear that these numbers are not comparable - they just serve to provide a rough impression on the range of accuracies reached. Further, we added Table A9 to evaluate the combination of our approach (NIFF) with further operations in the frequency domain. Thus we show that we can include the downsampling via FLC Pooling [3], complex BatchNorm [4] and also global average pooling plus the fully connected layer with NIFF to train reasonably well-performing models. However, we find that, to keep up accuracy, we need to return to the spatial domain for the non-linearity.
>
> 2. Number of hyperparameters, training and inference times:
>
> We now report the number of hyperparameters for our networks trained on ImageNet-1k and ImageNet-100 in Table 2 and for CIFAR-10 in Table A2. In terms of training times we report the average training time per epoch for CIFAR-10 and ImageNet-100 in Tables A5 and A6. We also added a similar evaluation for the inference time in Tables A7 and A8.
>
> 3. Training time:
>
> We report the training times on CIFAR-10 and ImageNet-100 in Tables A5 and A6. We show that our NIFF is faster than large convolutions, yet, slower than the small convolutions normally applied in CNNs.
>
> a) Refinement of Figures 2 and 3:
>
> We modified Figures 2 and 3 to provide more clarity (see above). Further, we distinguish our approach from using the naive implementation of neural implicit function based on single coordinates and how we address these challenges with NIFF.
>
> b) Performance comparison:
>
> We evaluate our methods on a single seed with the standard hyperparameters. Unfortunately, a thorough hyperparameter optimization for all models on ImageNet-1k, which is most likely needed to produce results on-par or better with the baseline, exceeds our computational resources.
>
> c) Duplicate references:
>
> Thank you for pointing this out, we fixed it.
>
> [1] Rao, Yongming, et al. "Global filter networks for image classification." Advances in neural information processing systems 34 (2021): 980-993.
>
> [2] Liu, Zhuang, et al. "A convnet for the 2020s." Proceedings of the IEEE/CVF conference on computer vision and pattern recognition. 2022.
>
> [3] Grabinski, Julia, et al. "Frequencylowcut pooling-plug and play against catastrophic overfitting." European Conference on Computer Vision. Cham: Springer Nature Switzerland, 2022.
>
> [4] Chiheb Trabelsi, et al. “Deep complex networks.” In International Conference on Learning Representations, 2018

---

> > ### Comment · Reviewer_Mxbe · 2024-04-01
> > **Follow-up from Reviewer**
> >
> > I thank the authors for their response and revision of the submission. The majority of my concerns have been addressed. I thank the author for being honest about the performance comparison to standard ConvNets trained with the timm recipe. While the performance of NIFF does not always outperform standard ConvNets, I think this is okay. It is also good to know that NIFF can be faster than large convolutions, showing its practical usage.
> >
> > One thing that still confused me it that in Table 2, the performance of DenseNet-121 Huang et al. (2017) (Pytorch) outperforms
> > DenseNet-121 Huang et al. (2017), which is weird. Shouldn't the latter outperform the former given its advanced training recipe?
> >
> > Overall, I am satisfied with the rebuttal.

---

> > > ### Author Response · Authors · 2024-04-03
> > >
> > > We thank the reviewer for the positive feedback on our revision! In the following, we will discuss the remaining question on DenseNet-121.
> > >
> > > We train with the training hyperparameters reported by the original work, assuming that in most cases, these are well-optimized for the respective models. The advanced training recipe only changes the data preprocessing. It does not necessarily mean that the combination of better preprocessing with the formerly optimized hyperparameters is still optimal. To achieve better performance, one might need to adapt the training hyperparameters to achieve higher performance with the advanced data preprocessing from [2]. Therefore, we are reporting both, the results achieved by timm and the results we achieve when training all models with the same preprocessing for all models for fair comparison to ours. Tuning training hyperparameters for each setup is beyond our compute resources, unfortunately.
> > >
> > > Please let us know if there are any further points you would like us to elaborate on!

---

### Review · Reviewer_wtTg · 2024-03-09

**Summary Of Contributions:**

This paper presents a novel approach which enables CNNs to efficiently learn infinitely large convolutional filters. Specifically, this paper introduces MLP parameterized Neural Implicit Frequency Filters (NIFFs) which learn filter representations directly in the frequency domain and can be plugged into any CNN architecture. Experiments are conducted on CIFAR and ImageNet dataset to demonstrate the effectiveness of the proposed method.

**Audience:**

Yes

**Claims And Evidence:**

Yes

**Requested Changes:**

See the weaknesses part

**Strengths And Weaknesses:**

### Strength

* The PCA analysis shown in Fig. 1 is interesting. The observation that large kernel networks still learn well-localized small kernels is useful for network design.
* The proposed method shows large improvements over ResNet, as shown in Table 2.

---
### Weakness

* Figure 2 needs to be carefully refined. It is quite difficult to capture the main idea of the proposed method in current form.
* The novelty and contribution of the proposed method is a bit unclear, since using FFT+IFFT is not a new idea in neural networks (GFNet etc.). More discussions and comparisons should be provided.
* As shown in Table 2, the proposed method suffers from 2% accuracy drop for DenseNet.
* The proposed method also does not perform well for mobile networks such as ConvNeXt-Tiny and MobileNet-v2. Besides, more modern mobile networks should be included rather than MobileNet-v2.

---

> ### Author Response · Authors · 2024-03-23
> **Answer to your Review**
>
> 1. Refinement of Figure 2:
>
> We refined Figure 2  to provide more clarity on the issue of naively increasing the kernel size and the proposed solution. Further, we distinguish our approach from using the naive implementation of neural implicit functions based on single coordinates and how we address these challenges with our NIFF.
>
> 2. Comparison to previous work:
>
> We compared our approach against previous approaches in Table 1 and distinguish more clearly which components operate in the frequency domain in each approach. In general, all previously proposed methods use specifically designed network architectures to include the convolution in the frequency domain. In comparison, our approach is a plug-and-play approach which can be used in any architecture to replace the convolution operation without increasing the number of trainable parameters.
>
> 3. Performance comparison:
>
> As already stated in our paper “Further, we want to emphasize that our NIFF is not conceived to improve over baseline methods (...) but to facilitate the evaluation of effective kernel sizes in CNNs”, we do not aim for better accuracy for our model. Further, we only evaluated one run on one random seed while not optimizing the hyperparameters at all due to computational constraints. Overall, the accuracies reached by NIFF are comparable to the ones of baseline models, so that they allow the study of filter sizes appropriately.
>
> 4. Additional network architecture:
>
> In addition to the six different models we evaluated in the original submission, we added MobileNet-v3 which is an optimized version of MobileNet-v2. The results are reported in Tables A1 and A2. Yet, we observe that both the baseline model and the NIFF version have comparably low accuracy. For MobileNets, the training pipeline is usually highly optimized for best performance. The data augmentation scheme from [1], which we employ for all trainings to achieve comparable results, does not seem to have a beneficial effect here, neither on ImageNet-1k (Table 2) nor on ImageNet-100 ( Table A1).
>
> [1] Liu, Zhuang, et al. "A convnet for the 2020s." Proceedings of the IEEE/CVF conference on computer vision and pattern recognition. 2022.

---

### Author Response · Authors · 2024-03-23
**Revision of our Manuscript**

We would like to thank all reviewers for their valuable and overall positive feedback! We are in particular
glad that they find that our "results are very appealing and encouraging for future research." - Reviewer
4rXV, and that our "proposed approach has great potential to reduce the complexity of infinite kernels."
-Reviewer Mxbe.

We are also glad for the feedback on improving our manuscript. We tried to incorporate these suggestions
as well as it was possible for us, given our available (academic) compute resources. Only a few of the suggested evaluations are missing in the current revision (e.g. a DenseNet-121 baseline models with depth-wise separated convolutions in Table 2), which we will add as soon as our computations finish.

An overview of our changes is listed in the "Changes Since Last Submission".
The parts that have been changed in the manuscript are highlighted in orange and figures as well as tables
that have been added are indicated as such by orange frames.

---

### Decision · Action_Editor_gZU5 · 2024-04-22

**Recommendation:** Accept as is

**Comment:**

The AE sides with acceptance in accordance with the thorough and detailed reviews on the technical content, empirical results, and overall claims and evidence in the submission. While one official recommendation leaned toward rejection, there is nevertheless consensus among two of the three reviewers and the AE. The argument for rejection is a complaint that fourier domain processing already exists in deep learning ("FFT+IFFT is not a new idea in neural networks (GFNet etc.)") and that certain convolutional architectures do not perform well with the method ("does not perform well for mobile networks such as ConvNeXt-Tiny and MobileNet-v2"), as well as a drop in accuracy for DenseNet. This argument does not engage with the full scope of the submission, or its chosen claims about exploring and applying adaptive and large kernel sizes, which are well-connected to the evidence.

The featured certification is recommended by the AE to (1) second the recommendation for this certification by a reviewer and (2) to reflect the satisfaction—enthusiasm even—by two of the three reviewers for the topic and execution of this work. It remains important to study fundamental factors of deep learning for vision, such as the degree of locality / the size of kernels, along with the pursuit of ever better general metrics like accuracy and efficiency.

The level of detail and background for each review offers more context for the decision. All three reviewers are experts on deep learning and vision with particular expertise in convolutional and attentional architectures, specialized filtering, and searching or optimizing over architectures/parameterizations for accuracy and generality. The reviewers Mxbe and 4rXV are the most positive and have the most expertise on filtering, the design and optimization of architectures for different scopes/resolutions/sparsity,  and implicit modeling. To be clear, the review and perspective of wtTg focusing on convnet architectures and baselines is also informative, and the AE likewise appreciates their service. The decision depends on the depth and breadth of each review, and the AE's own background on learning and filtering, and so to summarize the recommendations were weighted by their level of detail and correspondence with the full content of the submission.

The AE thanks the authors and reviewers for engaging in the TMLR process to deliver an improved paper with agreement between its claims and evidence and a clear audience.

**Audience:**

The reviewers and AE all agree that there is an audience at TMLR for this work. The audience includes researchers interested in (1) unifying or harmonizing signal processing and deep learning, (2) scale/resolution/locality as factors for architectural design in deep learning for vision, and (3) applications of implicit modeling as parameterization.

The AE would like to suggest an additional audience and topic of related work for the authors to consider: dynamic filtering/steerability/learned filter sizes. For instance, [deformable convolution](https://arxiv.org/abs/1703.06211), [resolution learning using scale space](https://arxiv.org/abs/2106.03412), [scale-equivariant steerable networks](https://arxiv.org/abs/1910.11093), and [deep scale-spaces](https://arxiv.org/abs/1905.11697) all address size and scale though only deformable convolution is in principle unbounded like the proposed NIFF (and still likewise has a practical limit). This is a suggestion for further visibility and impact and not a requirement for a revision.

**Claims And Evidence:**

The reviewers and AE all agree that the claims and evidence are connected in this work. The desiderata for the method and experiments are signaled well by the abstract (i-iv) and each point is examined and justified by the content.

The only point of feedback the AE feels compelled to mention is to suggest mild tempering of the last line of the abstract. The "best-case" filter sizes here are those for this particular parameterization through implicit frequency representations of filters; future work could arrive at other parameterizations with other optimal sizes.

---

> ### Author Response · Authors · 2024-05-14
> **Thank you**
>
> We would like to thank all reviewers and the AE for their valuable feedback and support for our work. We appreciate the time and effort put into the reviews and discussion of our work and are highly honoured by the certificate granted!
>
> Thank you again and best regards,
>
> the authors